

# How to determine the effective discharge and its return period in a semi-arid basin? The case of the Wadi Sebdou, Algeria (1973-2004)

Abdesselam Megnounif[1], Sylvain Ouillon[2]

[1]Laboratoire EOLE, Département d'hydraulique, Faculté de Technologie, Université Aboubekr Belkaid, Tlemcen, Algeria
[2]LEGOS, Univ. of Toulouse, IRD, CNRS, CNES, Toulouse, France

*Correspondence to*: Sylvain Ouillon (sylvain.ouillon@ird.fr)

**Abstract.** Over a long multi-year period, flood events can be classified according to their effectiveness in moving sediments. Efficiency depends both on the magnitude and frequency with which events occur. In this study, the efficiency of the Wadi Sebdou (North-West Algeria), in a semi-arid environment, is examined through its histogram of sediment supply by discharge classes, established from 31-years of measurements. The effective (or dominant) discharge is the one whose class corresponds to the maximum sediment supply. Three types of subdivisions into discharge classes were compared. The subdivision in classes of equal amplitudes and the subdivision with equivalent discharges were those which allowed a correct distribution of frequencies and supplies of water and sediments. The effective discharges for these two subdivisions were close and almost equal to the "half load discharge", i.e. to the flow rate corresponding to 50% of the cumulative sediment yield. The substitution of the flow histogram by a probability relationship and the use of a sediment rating curve enable to infer a theoretical value of the effective discharge. In this basin with strongly irregular flows, the introduction of a probability distribution was tested and assessed, analytical solutions are provided, but the Log-normal and Log-Gumbel laws highly underestimated the effective discharge. Return periods, estimated from the annual series of maximum discharge and half-load discharge, were compared. The former gives the period between hydrological years with discharges higher than the effective discharge (around 2 years), and the latter shows that more than half of the yearly sediment supply is carried by flows higher than the effective discharge only every 7 hydrological years. The study finally emphasized that the distribution of suspended load as a function of liquid discharge was sensitive to the basin and its forcings. On the Wadi Sebdou, the distribution of the sediment load, bimodal before 1988, became essentially monomodal after this date.

**Keywords:** sediment transport, effective discharge, dominant discharge, return period, semi-arid, Wadi

## 1 Introduction

Over a long multi-year period, flood events can be classified according to their effectiveness in moving sediments. Efficiency depends both on the magnitude and frequency with which events occur. According to Wolman & Miller (1960), the efficiency can be examined through the 'sediment transport effectiveness curve' h(Q) obtained by the product of the two curves f(Q).g(Q), where f(Q) is the frequency distribution of water discharge, and g(Q) the rating curve estimating the suspended sediment flux $Q_s$ as a function of the water discharge (Fig. 1). Since f(Q) is a bell-shaped probability density function, often adjusted by a





log-normal probability distribution, and g(Q) a function limited in the interval [0; Qmax[ then the function h(Q) goes from 0 at very low flow rates to almost 0 at the highest flow rates through a maximum. The flow at which the function h reaches its maximum is the effective discharge, $Q_D$, in the sense of Wolman & Miller (1960). The curve h(Q) characterizes the relative geomorphic work (i.e. the amount of sediment transported) that is carried out in a basin by each flow. The effective discharge

is often referred to as the dominant discharge, which has the greatest role in the formation and maintenance of river morphology, and whose knowledge is essential for stream restoration projects (Watson et al., 1999). As illustrated in Fig. 1, a large portion of sediment is conceptually transported by weak to moderate floods. Wolman & Miller (1960) confirmed this concept by comparing the frequency of flows generating suspended sediment transport on watersheds of different sizes in humid and semi-arid regions, and showed that very large devastating floods which produce large amounts of sediments have,

due to their scarcity, a tiny contribution over a long period in front of moderate floods with higher recurrence.

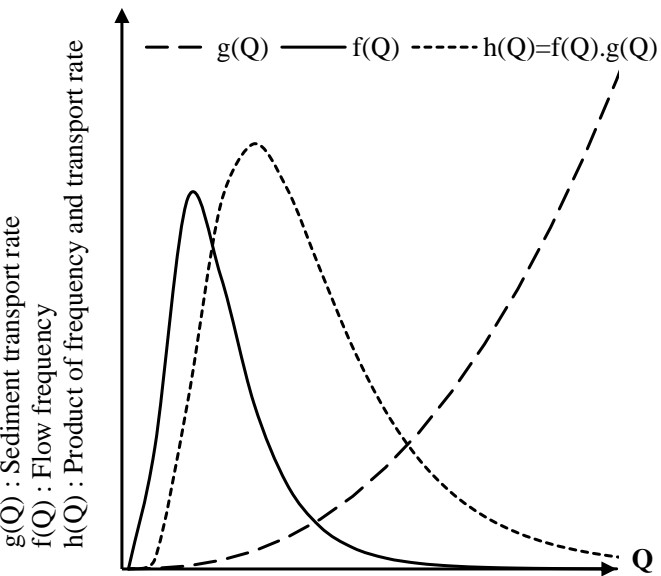

**Figure 1: Effective discharge curves.**

To determine the effective discharge, Benson and Thomas (1966) subdivided the discharge axis into classes of equal amplitude and replaced the sediment-transport effectiveness function, h(Q), with a sediment-load histogram. They defined the modal

class as the efficient-flow class or dominant class (Dunne and Leopold, 1978). When series of measurements spans several years, theoretical frequencies are deduced from the frequency distribution of measured discharges (Andrews, 1980). This approach made it possible to identify the dominant class on many sites, usually from daily liquid and solid flow series (Ashmore and Day, 1988; Biedenharn et al., 2001).

The simplest and most straightforward approach to estimating the class at effective discharge was summarized by Crowder

and Knapp (2005) in 5 steps:





- The flow series is subdivided into a number of equally spaced classes called 'intervals of stream discharge' (Benson and Thomas, 1966) or 'class intervals' (Biedenharn et al., 2001).

- The mean daily sediment load of a class is the average of the daily suspended solid loads conveyed by the discharges within the class considered.When the daily fluid flow rates are known but the solid load has not been measured, the solid discharge is estimated using a C-Q or $Q_s$-Q rating curve established on the considered station from daily values.

- The flow duration (in days) is estimated by summing the number of mean daily discharge values occurring within each discharge interval.

- The total sediment load transported by each liquid flow class is estimated by multiplying the class interval's average sediment load by the flow duration of that class interval.

- Lastly, a histogram of the sediment supplies is plotted against the discharge classes. The modal class, that carries the greatest amount of sediment, is the 'dominant class' and the mode is the 'effective discharge'.

In many rivers where flow variation is slow, water sampling required for solid flow measurement is not carried out daily but at monthly or weekly intervals (Horowitz, 2003). In this case, daily solid discharge is estimated by interpolation between actual measurements. On the other hand, small drainage basins (less than 1000 km$^2$) experiencing high intensity rainfall can generate short floods with high variation where recession sometimes lasts less than 24 hours. In these basins, monitoring sediment concentration requires a measurement protocol with a suitable, more tightened, temporal resolution. For a small alpine catchment river, Lenzi et al. (2006) adapted the Crowder and Knapp (2005) approach to hydrometric data at 5-minute intervals (the sediment concentration was deduced from water samples taken by automatic equipment at 5-minute intervals). Their results suggested that there are two dominant flow classes for mountain rivers: a relatively frequent discharge class responsible for maintaining the channel shape, and another class of high and infrequent flows that are responsible for macro-scale channel form.

Regarding the choice of discharge classes, the procedure is empirical and varies according to the authors (Pickup and Warner, 1976; Andrews, 1980; Lenzi et al., 2006). Biedenharn et al. (2001) recommended starting by the use of 25 classes of equal lengths. If no measurement is assigned to a class interval or the mode is isolated in the last histogram class corresponding to the highest rates, the number of classes is changed. Crowder and Knapp (2005) argued that each class must contain at least one flow of a flood event. Thus, this procedure is subjective and remains dependent on the measurement protocol and the watershed configuration (Sichingabula, 1999; Goodwin, 2004). For example, Hey (1997) showed that it is necessary to increase the number of classes to 250 for a suitable representation of the distribution of the sediment yield brought by the Little Missouri River at Marmarth and Medora. Yevjevich (1972) suggested that the number of classes should be between 10 and 25, depending on the size of the sample. He proposed that the length of the class interval does not exceed s/4, where s is the standard deviation of the series of studied liquid flows.

In addition to these methods, which only require the time series of Q and C, Nash (1994) proposed an analytical solution to estimate the effective discharge. He argued that for most rivers, the log-normal distribution is robust and adequately represents the flow frequency, and that sediment supply is commonly estimated from a power model, g(Q)= $aQ^{b+1}$, where a and b are



empirical parameters of simple regression established between the $C^k$ and $Q^k$ class representatives (Andrews, 1980; Biedenharn et al., 2001; McKee et al., 2002; Crowder and Knapp, 2005; Bunte et al., 2014). The probability distribution of discharges and the rating curve of sediment supply provide a mathematical equation of the sediment transport efficiency curve, h(Q) (Nash, 2004; Vogel et al., 2003). The curve h has a unique maximum reached at the effective rate $Q_D$ (fig.1), whose analytical

expression is solution of the derived function, h'(Q)=0. For more precision and in order to deal with different flow regimes, the analytical solution of the dominant discharge has been established for probability distributions other than the log-normal distribution, such as the normal, exponential or log-Pearson III distributions (Goodwin 2004; West and Niezgoda 2006; Quader et al., 2008; Higgins et al. 2015).

Other methods are still proposed in the literature to estimate the effective discharge. Ferro and Porto (2012), for example,

associated it to the flow rate corresponding to 50% of the cumulative sediment yield, thus taking up the concept of "half-load discharge" introduced by Vogel et al. (2003). Since flows below this threshold carry 50% of the total sediment production and higher flow rates as much, this flow can also be called a "median discharge in the sense of sediment yield" ($Q_{Y1/2}$). Other parameters are calculated in the literature and considered as proxies of the effective discharge, such as the bankfull discharge ($Q_b$, the discharge which fills the channel to the level of the floodplain, see e.g. Andrews, 1980) or the 1.5 years flow events

($Q_{1.5}$) (e.g. Crowder and Knapp, 2005; Ferro and Porto, 2012).

These approaches to analyze sediment yield are less well adapted to semi-arid environments that experience the alternation of very long periods of drought or low flows and sporadic floods, as is the case in northern Algeria. In the context of current knowledge and methods, this article proposes to examine these methods on an example and to compare the sediment load histograms resulting from three types of subdivision of discharge classes. The application is carried out from 31 years of hydro-

sedimentary measurements on the Wadi Sebdou (1973-2004), on which floods last on average 7.78% of the time. Nash's (1994) method of fitting statistical probability distributions to flow histograms to derive the dominant discharge is also applied to this basin. Analytical solutions are established for two standard probability distributions (log-normal, log-Gumbel). Their results are discussed in relation to the dominant discharges deduced from the sediment load histogram and different sources of error are examined.

**2 Study area**

The Maghreb is a mountainous region with young relief, characterized by many small watersheds. In these steep marl landscapes, rainfall erosivity is particularly high (Heusch, 1982; Probst and Amiotte-Suchet, 1992). Located in the north-west of Algeria, the Wadi Sebdou (or upper Tafna River) runs along 29-km. The upper reaches emerge through predominantly carbonate Jurassic terrains at altitudes up to 1400 m. Then the wadi crosses the plain of Sebdou composed of Plio-quaternary

alluviums, and a valley (the gap of Tafna) made up of carbonate rocks (marl-limestone, limestone and Jurassic dolomites) (Benest, 1972; Benest et al., 1992). The Wadi Sebdou flows into the Beni-Bahdel reservoir, with a storage capacity of 63 million m³, impounded in 1946. The Wadi Sebdou drainage basin area is 256 km². Steep slopes exceeding 25% represent about



49% of the total basin surface. The climate is semi-arid. The wet season runs from October to May. The dry season runs from June to September with low rainfall and high evapotranspiration. The main information on morphometric, geological and land use characteristics of the basin are reported in Bouanani (2004), Megnounif et al. (2013), and Megnounif and Ghenim (2016).

## 3 Methodology

### 3.1 Elementary contributions and budgets

The suspended sediment concentration is estimated from a water sample taken from the streambank. The product of discharge, Q in $m^3 \, s^{-1}$, and suspended sediment concentration, C in $g \, L^{-1}$ (or $kg \, m^3$), make it possible to evaluate the sediment yield, $Q_S = Q.C$ in $kg \, s^{-1}$. Between two water samples, the liquid flow, Q, and the sediment discharge $Q_S$, are assumed to vary linearly. At each flow $Q_i$ measured at time $t_i$, is associated a triplet ($\Delta t_i$, $\Delta R_i$, $\Delta Y_i$):

$$\Delta t_i = \frac{1}{2}(t_{i+1} - t_i) + \frac{1}{2}(t_i - t_{i-1}) = \frac{1}{2}(t_{i+1} - t_{i-1}), \tag{1}$$

$$\Delta R_i = \frac{1}{4}[(Q_{i+1} + Q_i)(t_{i+1} - t_i) + (Q_i + Q_{i-1})(t_i - t_{i-1})]10^{-6}, \tag{2}$$

$$\Delta Y_i = \frac{1}{4}[(Q_{i+1}C_{i+1} + Q_iC_i)(t_{i+1} - t_i) + (Q_iC_i + Q_{i-1}C_{i-1})(t_i - t_{i-1})]10^{-6}, \tag{3}$$

where $\Delta t_i$, $\Delta R_i$ and $\Delta Y_i$ correspond to time duration (in s) and elementary inputs in water (unit: $10^6 \, m^3$) and sediment load (unit: $10^3$ tonnes) assigned to the discharge $Q_i$, respectively.

Over a duration T, the water contribution $R_T$ and solid contribution $Y_T$ are estimated by summing the elementary contributions:

$$R_T = \sum_{t_i \in T} \Delta R_i \qquad Y_T = \sum_{t_i \in T} \Delta Y_i, \tag{4}$$

Various quantiles are given using cumulative frequencies and elementary contributions assigned to ordered discharges. The quantiles $Q_{T\alpha}$, $Q_{R\alpha}$ and $Q_{Y\alpha}$ stand for discharges that delimit α% of annual time, α% of the total water supply and α% of the sediment yield, respectively. For example, $Q_{Y1/2}$ is the median discharge in terms of suspended sediment production, i.e. such that 50% of the sediment production is carried by discharges lower than $Q_{Y1/2}$.

### 3.2 Class interval assignment and dominant discharge

#### 3.2.1 Class intervals and associated parameters

To analyze flow frequencies and associated sediment supplies, the x-axis (discharge) is subdivided into class intervals $I_k = [a_k, a_{k+1}[$ where $k = 0, ..., N$ ; $a_0 \leq Q_{min} < a_1$ and $a_N \leq Q_{max} < a_{N+1}$. The duration, water and sediment supplies attributed to the class $I_k$ are obtained by summing time intervals (eq. 1), water contributions (eq. 2) and sediment contributions (eq. 3) assigned to discharges $Q_i$ within this class, following:

$$\Delta T^k = \sum_{Q_i \in I_k} \Delta T_i \; ; \; \Delta R^k = \sum_{Q_i \in I_k} \Delta R_i \; \text{and} \; \Delta Y^k = \sum_{Q_i \in I_k} \Delta Y_i \; . \tag{5}$$

Each discharge class, $I_k$, is represented by a pair ($Q^k$, $C^k$) where $Q^k$, is the midpoint and $C^k$ is the mean sediment concentration calculated by the equation:





$$C^k = \frac{\Delta Y^k}{\Delta R^k} \qquad (6)$$

Discharge classes are examined according to their effectiveness to produce sediments. The discharge class that carries out the highest sediment load over an extended period is the dominant class. For simplicity, the midpoint, $Q_D$, is the effective discharge or dominant discharge. In this article, three ways to subdivide the discharge-axis are presented, applied and compared.

### 3.2.2 Classes of equal length

The series of ordinal discharges is subdivided into class intervals of equal size. A flow frequency and percentages of water and sediment contributions are assigned to each class interval. Various class lengths are examined and compared to that of length $1 \, m^3 \, s^{-1}$.

### 3.2.3 Classes of equal water supply

Based on a physical aspect, the second subdivision ensure that classes provide the same water supply. For that, cumulative frequencies and water and sediment elementary supplies in percentage ($\sum_{j \leq i} \Delta T_i\%$, $\sum_{j \leq i} \Delta R_i\%$, $\sum_{j \leq i} \Delta Y_i\%$) are assigned to the ordinal discharges. Class boundaries are delimited according to the cumulative water supply. For example, to get 25 classes, elementary water inputs assigned to each class must accumulate 4% of the total water supply. At equal water yields, the efficient class is the one that carries out the most sediments.

### 3.2.4 Classes in geometric progression

Initiation of sediment motion by water depends on shear stress (Shields, 1936). In many sediment transport models, the sediment transport rate per unit channel width ($q_s$) follows a power law as a function of excess shear stress $q_s = k \, (\tau - \tau_c)^n$ where $\tau$ is the shear stress per unit area and $\tau_c$ is the critical stress of sediment required for grain motion, k is a parameter depending on sediment particle characteristics, and *n* is an empirical exponent (e.g. Bagnold, 1941; van Rijn, 2005). As a result, power law models are commonly used, where sediment discharge $Q_S$ or sediment concentration C evolves as a function of water discharge Q (Walling 1977):

$$Q_S = aQ^{b+1} \quad or \quad C = aQ^b \qquad (7)$$

or, in a consistent manner:

$$\log C = \log a + b \, \log Q \qquad (8)$$

In a stream that verifies such relationship, the sediment discharge varies linearly against the water discharge on a logarithmic scale. For this reason, we suggest subdividing the x-axis (discharge) into classes of equal lengths on a logarithmic scale. Hence, class limits ($a_i$) are chosen so that:

$$\log a_{i+1} - \log a_i = \beta \, (Constant) \qquad (9)$$

Since the log function is bijective on $R^+$ (positive real numbers), for a constant $\beta > 0$, there exists $\alpha > 0$ such that $\beta = \log(1+\alpha)$. Whence:



$$\log\frac{a_{i+1}}{a_i} = \log(1 + \alpha) \iff a_{i+1} = a_i(1 + \alpha) \tag{10}$$

In this case, the length of classes is in a geometric progression of common ratio $1+\alpha$ and all the class limits may be deduced from $a_0$, according to:

$$a_{k+1} = a_k(1 + \alpha) = a_0(1 + \alpha)^k \tag{11}$$

For a small value of $\alpha$, appropriately chosen, discharges within each class can be considered as equivalent to the value at the center of the class, $a^k = a_k\left(1 + \frac{\alpha}{2}\right)$, since:

$$\forall Q \in [a_k\, ; a_{k+1}[\, ; \ \frac{(Q - a^k)}{a^k} \le \frac{\alpha}{2} \tag{12}$$

The sediment supply assigned to each discharge class is represented by a histogram on logarithmic scale or by a bar graph on arithmetic scale. The midpoint of the modal class interval represents the effective discharge $Q_D$.

## 3.3 Hydrometric measurements and data pre-processing

Discharge and concentration data were measured at the Beni Bahdel station by the National Agency of Hydraulic Resources (locally called ANRH, www.anrh.dz), in charge of gauging stations and measurements in Algeria. These data cover a 31-year period from September 1973 to August 2004. When water level is low and stable, the operator takes water samples every other day. During flood periods, sampling is intensified, up to every half hour. During low flow period, water samples are taken every two weeks. At each sampling, the operator reads the water level on a limnimetric scale or on a limnigraph which is then converted into a water discharge according to a locally made abacus. The suspended sediment concentration is determined from the sampled water after filtration, following a protocol described in Megnounif et al. (2013). The measurement protocol of the ANRH services is based on a predefined calendar. However, the high variability of the flows experienced by the Wadi Sebdou is such that between two consecutive measurements the difference can be significant and one class or more may not be represented by any flow, whatever the subdivision used to discretize the flow discharges into classes. Moreover, such large differences cause an overestimate of the contributions in the sampled classes and underestimate those that are not.

According to Horowitz (2003), the vast majority of suspended sediment concentration data result from manually collected individual samples. He pointed out that very few sites are equipped with automatic equipment (automatic pump for sampling water, or turbidimeters) to provide continuous or near-continuous records of suspended sediment concentration data. In addition to its excessive cost, this equipment may be damaged or washed away by torrential flows (Alexandrov et al., 2007). Colombani et al. (1984) and Castillo et al. (2003) emphasized practical difficulties to control flows and associated matters in small catchments (10 to $10^4$ km$^2$), like the Wadi Sebdou. Such catchments are subject to flash floods that carry significant sediment loads (Reid and Laronne, 1995; Alexandrov et al., 2003; Scott, 2006; Gray et al., 2015) and where accurate sediment records are frequently lacking (Milliman and Syvistki 1992; Biedenharn et al., 2001; Gray et al. 2015). Probst and Amiotte-Suchet (1992) and Walling (2008) reported that the lack of such series is obvious on the southern Mediterranean side. Due to the paucity of accurate time series, Crowder and Knapp (2005) highlighted that the approach developed for identifying the



effective discharge has not been verified in watersheds smaller than 518 km$^2$. Biedenharn et al. (2001) and Gray et al. (2015) reported that, in small basins with irregular flow, the identification of effective discharge is complex and requires a coverage of hydrometric measurements with a fine time resolution (less than one hour). According to Simon et al. (2004), the scarcity of such records in the U.S.A. makes difficult to identify the regional effective discharge.

A preliminary data processing was performed in this study in order to improve the distribution of elementary inputs amongst classes. To achieve this, liquid and solid discharges are assumed to vary linearly as a function of time between two measurements. When the discrepancy between two measured discharges is large, an intermediate discharge is added at each increase of 0.2 m$^3$ s$^{-1}$. The corresponding values of time and sediment discharge are deduced using linear interpolation between measurements. The value of 0.2 m$^3$ s$^{-1}$ was chosen close to the baseflow observed in the river, $Q_0 = 0.16$ m$^3$ s$^{-1}$ (Terfous et al.,

2001; Megnounif et al., 2003). This preliminary data treatment allows to better distribute the information amongst the classes and to estimate in a more continuous way the elementary inputs. Thus, the data series on which we applied and compared methods has increased from 6,947 initial measurements collected by the ANRH to 40,081 data ($t_i$, $Q_i$, $C_i$).

### 3.4 Relevance of a subdivision

The relevance of a subdivision was examined according to its ability to represent the water and sediment supplies. Three

aspects were considered:

- A subdivision was considered suitable when histograms were informative on the three variables (frequency, water supply and sediment supply) evolution over the whole flow range, from the weakest to the strongest.
- The water and sediment inputs assigned to each discharge class can be quantified by the 'standard' elementary contributions (Eq. 5), or alternatively estimated using the midpoint discharge and the mean sediment concentration

of each class (Eq. 6). Discrepancies are expressed as a percentage by the ratios $\tau_{Rk}$ and $\tau_{Yk}$, such as:

$$\tau_{Rk} = \frac{Q^k \Delta T^k 10^{-6} - \Delta R^k}{\Delta R^k} 100 \quad \text{and} \quad \tau_{Yk} = \frac{Q^k C^k \Delta T^k 10^{-6} - \Delta Y^k}{\Delta Y^k} 100 \tag{13}$$

When estimating total water and sediment supplies, discrepancies are given by:

$$\tau_R = \frac{\sum_{k=0}^{N} Q^k \Delta T^k 10^{-6} - \sum_{k=0}^{N} \Delta R^k}{\sum_{k=0}^{N} \Delta R^k} 100 \quad \text{and} \quad \tau_Y = \frac{\sum_{k=0}^{N} Q^k C^k \Delta T^k 10^{-6} - \sum_{k=0}^{N} \Delta Y^k}{\sum_{k=0}^{N} \Delta Y^k} 100 \tag{14}$$

A subdivision is better when it provides the smallest discrepancies according to the equations 13 and 14. Note that,

for the same class, the differences $\tau_{Rk}$ and $\tau_{Yk}$ are identical. Indeed, Eqs. 6 and 13 give:

$$\tau_{Yk} = \left( \frac{Q^k C^k \Delta T^k . 10^{-6} - \Delta Y^k}{\Delta Y^k} \right) . 100 = \left( \frac{Q^k \frac{\Delta Y^k}{\Delta R^k} \Delta T^k . 10^{-6} - \Delta Y^k}{\Delta Y^k} \right) . 100 \tag{15}$$

After simplification of the term $\Delta Y^k$, we find that: $\tau_{Yk} = \tau_{Rk}$.

- An additional criterion was considered to determine the effective discharge from analysis. The suspended sediment concentration assigned to each class $C^k$ may be alternatively estimated from the power model $C = a\,Q^b$ fitted with

class representatives ($Q^k$, $C^k$), a and b being empirically derived regression coefficients. A subdivision is relevant





when, on the one hand, the coefficient of determination and the coefficient of Nash and Sutcliffe between measured sediment loads and estimated values were close to one, and on the other hand, the subdivision yields the smallest differences between sediment load using Eq. 5 and sediment estimate using the power model:

$$\tau_{MYk} = \left( \frac{a(Q^k)^{b+1} T^k 10^{-6} - \Delta Y^k}{\Delta Y^k} \right) 100 \tag{16}$$

The total discrepancy was quantified by the ratio:

$$\tau_{MY} = \left( \frac{\sum_k a(Q^k)^{b+1} T^k 10^{-6} - \sum_k \Delta Y^k}{\sum_k \Delta Y^k} \right) 100 \tag{17}$$

### 3.5 Analytical determination of the effective discharge

Probability density functions representing flow frequencies are left skewed distributions. The most commonly used is the log-normal distribution (Wolman and Miller, 1960; Nash, 1994). However, for irregular flows, more pronounced asymmetric distributions are recommended. Hence, in addition to the log-normal distribution, the exponential and log-Gumbel distributions were examined. The theoretical density functions were fitted to the discharge frequency histogram. The dominant discharge was deduced from the analytical solution of h'(Q) = 0, using the sediment rating curve C-Q fitting the pairs $(Q^k, C^k)$ Analytical solutions for the log-normal and log-Gumbel distributions are given in detail in the following subsections. The relevance of these solutions was assessed through the ability of the sediment-transport effectiveness curve to represent the sediment load histogram, globally and within class intervals.

### 3.5.1 Effective discharge using a log-normal distribution

The 2-parameter log-normal distribution has a probability density function:

$$f(Q) = \frac{1}{\delta Q \sqrt{2\pi}} \exp\left[ -\frac{1}{2} \left( \frac{Ln(Q)-\mu}{\delta} \right)^2 \right] \tag{18}$$

where $\mu$ and $\delta$ are the mean and standard deviation of the Ln(Q) distribution. So, the sediment transport effectiveness curve can be written:

$$h(Q) = \frac{1}{\delta Q \sqrt{2\pi}} \exp\left[ -\frac{1}{2} \left( \frac{Ln(Q)-\mu}{\delta} \right)^2 \right] aQ^{b+1} \tag{19}$$

The derivative of the function h is given by:

$$h'(Q) = \frac{aQ^{b-1}}{\delta\sqrt{2\pi}} \exp\left[ -\frac{1}{2} \left( \frac{Ln(Q)-\mu}{\delta} \right)^2 \right] \left[ -\frac{Ln(Q)-\mu}{\delta^2} + b \right] \tag{20}$$

$h'(Q) = 0$ when $-\frac{Ln(Q)-\mu}{\delta^2} + b = 0$, and so:

$$Q_D = \exp(\mu + b\delta^2) \tag{21}$$

The mode is the discharge value that appears most often. It is the discharge at which the probability density function has a maximum value. The analytical solution of $f'(Q) = 0$ gives:

$$Q_{mode} = \exp(\mu - \delta^2) \tag{22}$$





### 3.5.2 Effective discharge using a log-Gumbel distribution

The two-parameter log-Gumbel distribution is defined through its probability density function:

$$f(Q) = -\exp(-u)'\exp(-exp(-u)) \quad \text{where} \quad u = a_g LnQ + b_g \tag{23}$$

for which the parameters, $a_g = \dfrac{\pi}{\delta\sqrt{6}}$ and $b_g = 0.5774 - a_g\mu$, are issued from the method of probability weighted moments, $\mu$

and $\delta$ being identical to the parameters of the log-normal distribution.

The function h is written:

$$h(Q) = -\exp(-u)'\exp(-\exp(-u))g(Q) \quad \text{with} \quad g(Q) = aQ^{b+1} \tag{24}$$

and its derivative h':

$$h'(Q) = -\frac{a_g}{Q^2}\exp(-u)'\exp(-\exp(-u))g(Q)\big[-a_g + a_g\exp(-u) + b\big] \tag{25}$$

Thus, the dominant discharge can be expressed by:

$$Q_D = \exp\left[-\frac{Ln\left(1-\frac{b}{a_g}\right)+b_g}{a_g}\right] \tag{26}$$

The solution of f'(Q)=0 gives the mode:

$$Q_{mode} = \exp\left[-\frac{Ln\left(1+\frac{1}{a_g}\right)+b_g}{a_g}\right] \tag{27}$$

### 3.6 Half-load discharge

In their study, based on 27 stream gauge stations located in three regions of southern Italy, Ferro and Porto (2012) liken the dominant discharge to the median discharge in terms of sediment yield ($Q_{Y1/2}$) , i.e., the discharge value above and below which half the long-term sediment load is transported. Vogel et al. (2003) previously introduced this parameter, which they called "half-load discharge" and that they distinguished from the effective discharge. The half-load discharge was determined for the Wadi Sebou by cumulating elementary sediment contributions assigned to the ordinal discharges covering the study

period 1973-2004. The obtained discharge, $Q_{Y1/2}$, was compared to the dominant discharge $Q_D$.

### 3.7 Recurrence interval

The series of hydrologic data, Q, employed to estimate the recurrence interval (or return period) of an event of a given magnitude $Q_p$, should be selected so that these values are independent and identically distributed along the considered time series. Such a series can compile any remarkable yearly discharge (e.g. average, maximum or minimum annual discharge).

Each year should have a unique representative value so that the number of base values equals the number of the study-years (Chow et al., 1988). The recurrence interval of an event of magnitude equal or exceeding $Q_p$ is $RI(Q_p) = \dfrac{1}{Prob(Q>Q_p)}$.




The estimate of the effective discharge recurrence interval is traditionally derived from the probability distribution fitted to the annual maximum discharge series (Biedenharn et al., 2001; Simon et al., 2004; Crowder and Knapp, 2005; Ferro and Porto, 2012; Bunte et al., 2014). However, this calculation relies only on hydrological measurements and does not consider the associated sediment supplies. In this work, the effective discharge recurrence interval estimated using the traditional approach

was compared to that estimated from probability distributions fitted to the annual half load discharge.

## 4 Results

### 4.1 Effective discharge values for various subdivisions

In the Wadi Sebdou, floods with high sediment concentrations are rare (Fig. 2). Discharges are greater than $Q_{T99} = 9.68$ m$^3$ s$^{-1}$ during 1% of the annual time, with an average sediment concentration being worth 10.3 g L$^{-1}$. They represent 25.0% of the

total water input and carry 82.8% of sediment. On the other hand, discharges lower than 1.54 m$^3$ s$^{-1}$ represent 90% of the annual time. Their weak average concentration is 0.19 g L$^{-1}$.

Interquartile discharges for water supplies, [0.66; 9.68 m$^3$ s$^{-1}$[, last 30.3% of the annual time and carry 15.6% of the total sediment load with an average concentration of 0.97 g L$^{-1}$. Discharges higher than $Q_{R99} = 85.2$ m$^3$ s$^{-1}$ have an average frequency of 0.01%, i.e. 1.3 hours per year. These waters carry 16.64% of total sediment production with an estimated average

concentration of 25.9 g L$^{-1}$.

The first and third quartiles for sediment production are delimited by $Q_{Y1/4} = 15.3$ m$^3$ s$^{-1}$ and $Q_{Y3/4} = 58.4$ m$^3$ s$^{-1}$. Approximately 4/5 (80.4%) of the total volume of water flows with discharges lower than the 1$^{st}$ quartile, with an average sediment concentration being worth 0.96 g L$^{-1}$. Discharges higher than the third quartile, heavy loaded with an average concentration of 15.4 g L$^{-1}$, account for only 5.1% of water supplies. They last 0.06% of the annual time, i.e. 5 hours per year, on average. The

half-load discharge $Q_{Y1/2} = 29.8$ m$^3$ s$^{-1}$ corresponds to 87.3% of water inputs that flows in 99.76% of the annual time, with an average concentration of 1.8 g L$^{-1}$. Over 131 floods recorded during the study period, 33 had a peak flow higher than the median flow, of which 11 were multi-peaks and exceeded 26 times the value of $Q_{Y1/2}$.

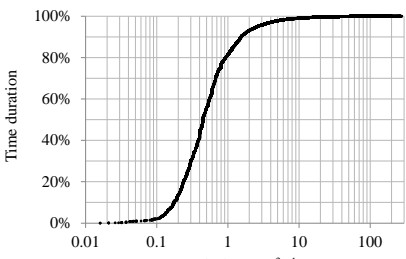
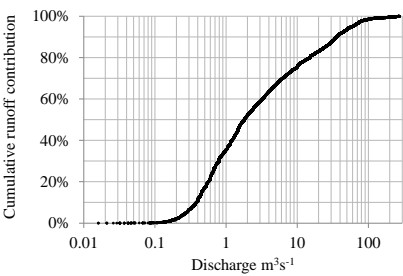
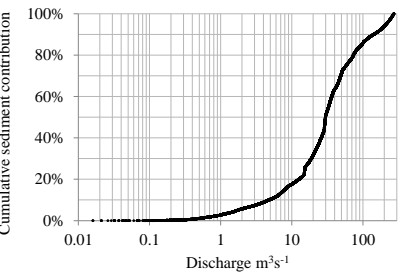

**Figure 2: Cumulative frequency, water and sediment inputs assigned to ordinal discharges in the Wadi Sebdou (1973-2004).**





### 4.1.1 Subdivision into classes of equal length

Discharges of the Wadi Sebdou discretized in classes of length equal to 1 m$^3$ s$^{-1}$ give 273 classes. The range of dominant discharge in terms of sediment production, [29; 30 m$^3$ s$^{-1}$[, corresponds to 4.8% of the total sediment supply (Fig. 3). This class represents 0.51% of the total water supply with an average concentration of 29.1 g L$^{-1}$ (Table 1), for a duration of 0.02%, i.e.

5  about 1.5 h per year, or 0.21% of the total flood duration which cover on average 7.78% of the year. The following classes, in order of efficiency to mobilize sediment, are: [15; 16[, [28; 29[, [1; 2[ and [0; 1 m$^3$ s$^{-1}$[ with sediment productions of 4.35, 3.05, 2.87 and 2.73%, respectively. The 2$^{nd}$ efficient class represents 1.16% of the total water input and lasts 0.073% of the time (around 6.4 hours per year). The classes with low discharges, [0; 1 m$^3$ s$^{-1}$[ and [1; 2 m$^3$ s$^{-1}$[, are the most frequent; they last 81.5% and 11.65% of the annual time, respectively, with average water inputs of 35.49 and 16.72%, respectively. Every

10  class above 38 m$^3$ s$^{-1}$ contribute for a sediment load of less than 1%. Their contribution decreases to less than 0.5% for discharges above 53 m$^3$ s$^{-1}$ (Fig. 3).

**Figure 3: Duration, water and sediment supplies, and sediment rating curve with a subdivision into classes of equal length 1 m$^3$ s$^{-1}$.**





For such a subdivision, a change in class length necessarily affects the representativeness of the flow characteristics, in particular the magnitude and position of the effective discharge $Q_D$. The latter varied from 29.5 to 25 m$^3$ s$^{-1}$ when the class length increased from 1 to 10 m$^3$ s$^{-1}$, i.e. when the number of classes was reduced from 273 to 28. The contribution of the dominant class changed accordingly, from 4.8 to 19.0% of sediment supply, and from 0.51 to 4.32% of water flow. The frequency of discharges in this class changed as well, from 0.02% to 0.17% of the annual time.

Table 1: Characteristics and performance of various subdivisions: class of dominant discharge range CD; effective discharge; flow frequency ΔT, water supply ΔR, sediment supply ΔY and concentration C; parameters of the rating curve $C^k = a Q^{k\,b}$; discrepancies between water and sediment inputs obtained from classes and from elementary contributions.

| | Classes of equal length | | | | Classes of equal water supply | | Classes in geometric progression |
| --- | --- | --- | --- | --- | --- | --- | --- |
| | 1 m$^3$ s$^{-1}$ | 2 m$^3$ s$^{-1}$ | 3 m$^3$ s$^{-1}$ | 4 m$^3$ s$^{-1}$ | (1%) | (4%) | (common ratio 1.2) |
| C.D. (m$^3$ s$^{-1}$) | 29-30 | 28-30 | 27-30 | 28-32 | 121-272.6 | 66.8-272.6 | 26.4-31.7 |
| $Q_D$ (m$^3$ s$^{-1}$) | 29.5 | 29.0 | 28.5 | 30.0 | 197.2 | 169.7 | 29.01 |
| ΔT (%) | 0.02 | 0.03 | 0.04 | 0.06 | 0.01 | 0.04 | 0.074 |
| ΔR (%) | 0.51 | 0.96 | 1.33 | 1.77 | 1.00 | 4.00 | 2.24 |
| ΔY (%) | 4.77 | 7.82 | 9.23 | 10.93 | 11.4 | 22.4 | 12.74 |
| C (g L$^{-1}$) | 29.08 | 25.5 | 21.6 | 19.3 | 35.2 | 17.5 | 17.8 |
| $\tau_R$ | 8.8% | 46.0% | 92.2% | 139.7% | -0.05% | 3.3% | 0.3% |
| $\min(\tau_{Rk})$ | -1.1% | -0.1% | -0.2% | -0.4% | -32.1% | -30.0% | -33.1% |
| $\max(\tau_{Rk})$ | 19.5% | 85.7% | 155.0% | 218.2% | 7.4% | 79.7% | 2.3% |
| a | 0.4874 | 0.4777 | 0.4539 | 0.4460 | 0.3876 | 0.4453 | 0.5032 |
| b | 0.8031 | 0.8072 | 0.8181 | 0.8213 | 0.8799 | 0.8138 | 0.7917 |
| R² | 0.879 | 0.879 | 0.879 | 0.878 | 0.906 | 0.946 | 0.950 |
| Nash-Suthcliffe | 0.888 | 0.890 | 0.897 | 0.898 | 0.769 | 0.588 | 0.930 |
| $\tau_{MY}$ | -6.0% | -1.2% | 14.2% | 32.5% | -6.0% | 31.8% | -6.9% |
| $\min(\tau_{MY^k})$ | -74.6% | -71.6% | -67.5% | -62.1% | -90.1% | -70.0% | -85.8% |
| $\max(\tau_{MY^k})$ | 109.1% | 160.4% | 313.8% | 474.1% | 487.4% | 199.0% | 102.2% |

The comparison between the water and sediment inputs estimated from class representatives ($Q^k$, $C^k$), on one side, and those directly calculated from elementary contributions, on the other side, shows that the subdivision into classes of length 1 m$^3$ s$^{-1}$ gives the smallest discrepancies, as calculated by equations 13 and 14. Deviations increase with increasing class sizes (Table 1). Similarly, power models, $C=aQ^b$, based on class representatives ($Q^k$, $C^k$) of equal lengths 1 and 2 m$^3$ s$^{-1}$ (Table 1) give the best rating curves: above 2 m$^3$ s$^{-1}$, the greater the amplitude, the higher the error in sediment production (see $\tau_{MY}$ in Table 1). Overall, with classes of equal amplitude, the informative part of histogram remains confined to low to moderate flow and





decreases when the class amplitude decreases (Fig. 4). However, an excessive increase or decrease in the length class, to more than 8 $m^3 s^{-1}$ or less than 0.5 $m^3 s^{-1}$, affects the quality of information on the flow efficiency and makes the reading of histograms not very informative.

**Figure 4: Sediment yields for subdivisions of equal lengths: 2, 4, 6 and 8 $m^3$ $s^{-1}$.**

### 4.1.2 Subdivision into classes of equal water supply

Subdivision into classes of equal water input of 4% includes 25 classes (Fig. 5). The choice of 4% allows to get 25 classes, as recommended by Biedenharn et al. (2001) and Crowder and Knapp (2005). The upper-class concerns discharges higher than 66.8 $m^3 s^{-1}$ and carries the most of sediment, accounting for 22.4% of the total annual sediment load. The frequency of concerned discharges is 0.04%. The effective discharge, at the center of the class, is $Q_D = 169.7$ $m^3 s^{-1}$. The second class in terms of efficiency, [22.1; 31.6 $m^3 s^{-1}$ [, carried 18.5% of the total sediment flow with a flow frequency of 0.14%. The last 5 highest classes, from 15 to 273 $m^3 s^{-1}$, collected 20% of water inputs and 80% of sediment inputs.





Although this subdivision describes a physical reality allowing a rather detailed reading of the frequency variations and water and sediment inputs at low flows, it remains basic and provides little details on the efficiency of moderate to high flows. The difference (eq.13) between direct calculation of water inputs (eq.2, 4) and the one based on discharges of each class (eq.5), although being globally low (3.3%), showed to be high for some classes (Table 1). The corresponding rating curve $C^k = a\,Q^k{}_b$

5   leads to underestimate by 32% the sediment load as compared to the elementary contributions (Table 1). The maximum gap (199%) was reached for the dominant class.

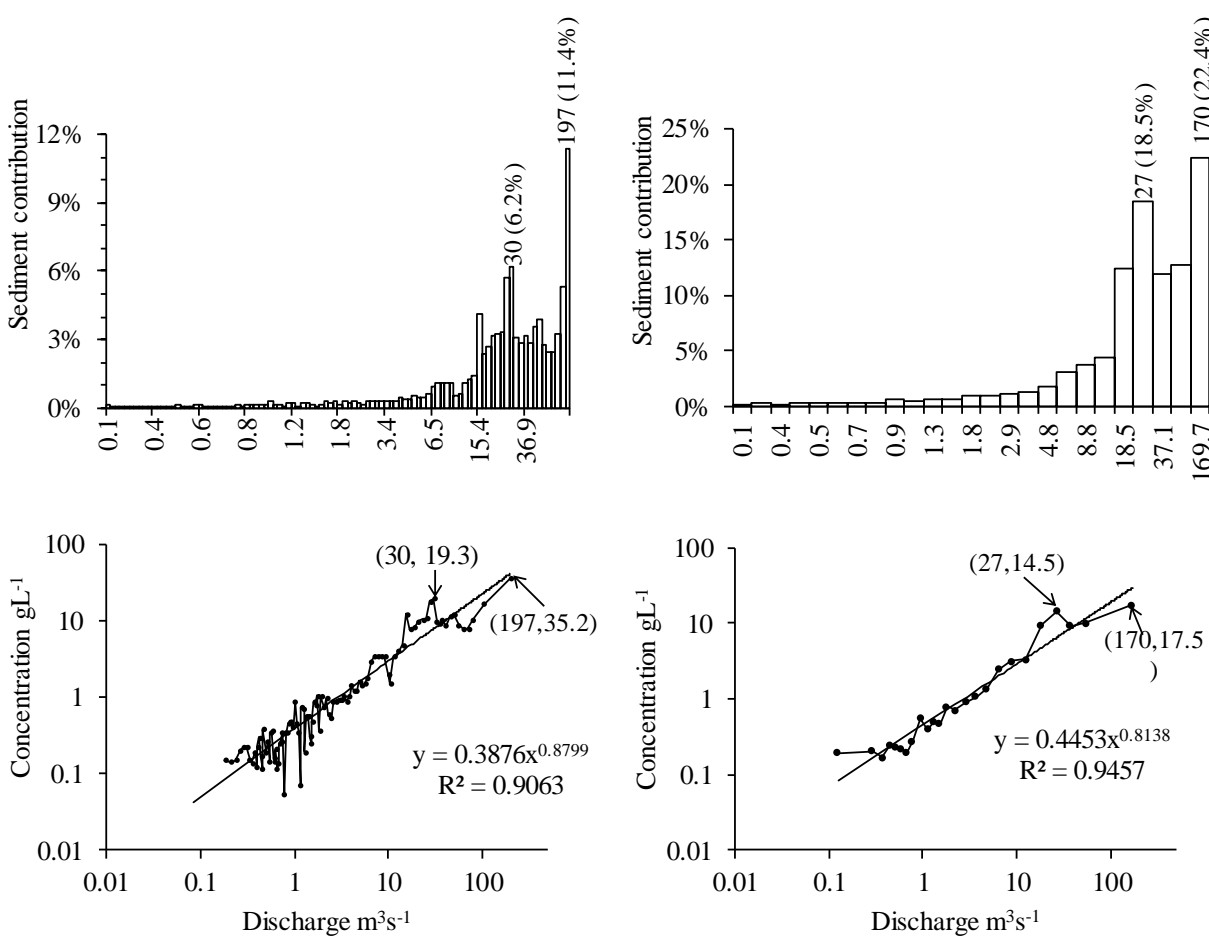

**Figure 5: Duration and sediment supply per class, and sediment rating curves, for a subdivision into classes of equal water supplies of 1% (left) and 4% (right).**

10   A calculation performed with a subdivision into 100 classes of equal water contributions of 1% (Fig. 5) reduced the errors made on $\tau_R$ and $\tau_{MY}$ (Table 1). However, despite a high coefficient of determination, differences class by class were too high and the maximum error, obtained on the last class which is the dominant class, was around 500%.



### 4.1.3 Subdivision into classes in geometric progression

The subdivision into classes of geometric progression was chosen so that from one class to another, the amplitude of the class increases by 20%. Thus, discharges in a same class are within 10% of the class center. In this case, on a logarithmic scale, classes have a length equal to $\beta = \log(1 + 0.2) \cong 0.0792$. Thus, the amplitude of classes is in geometric progression of common

ratio 1.2. The initial term $a_0 = Q_{R1\%} = 0.164$ m$^3$ s$^{-1}$ corresponds to the flow delimiting 1% of water supplies. This subdivision required 42 classes to cover the discharges of the Wadi Sebdou over 1973-2004. The class [26.4; 31.7 m$^3$ s$^{-1}$[ stands out and dominates with a relative contribution of 12.74% of the total sediment supply. Its average frequency is 0.074% or 6.5 h per year (Fig. 6). The water supply from this class represents 2.24% of the total, with an average concentration of 17.8 g L$^{-1}$. Bar graphs (Fig. 6) (or histograms on a logarithmic scale) allow a fairly detailed representation and reading of the flow frequency

distribution and of the water and sediment supplies, as well, for the different flow regimes. In addition, the difference between water and sediment supplies estimated from class representatives and those calculated from elementary contributions is almost nil in total, and is low to moderate with different classes (Table 1). The maximum error coincides with the 1$^{st}$ class assigned to low flows.

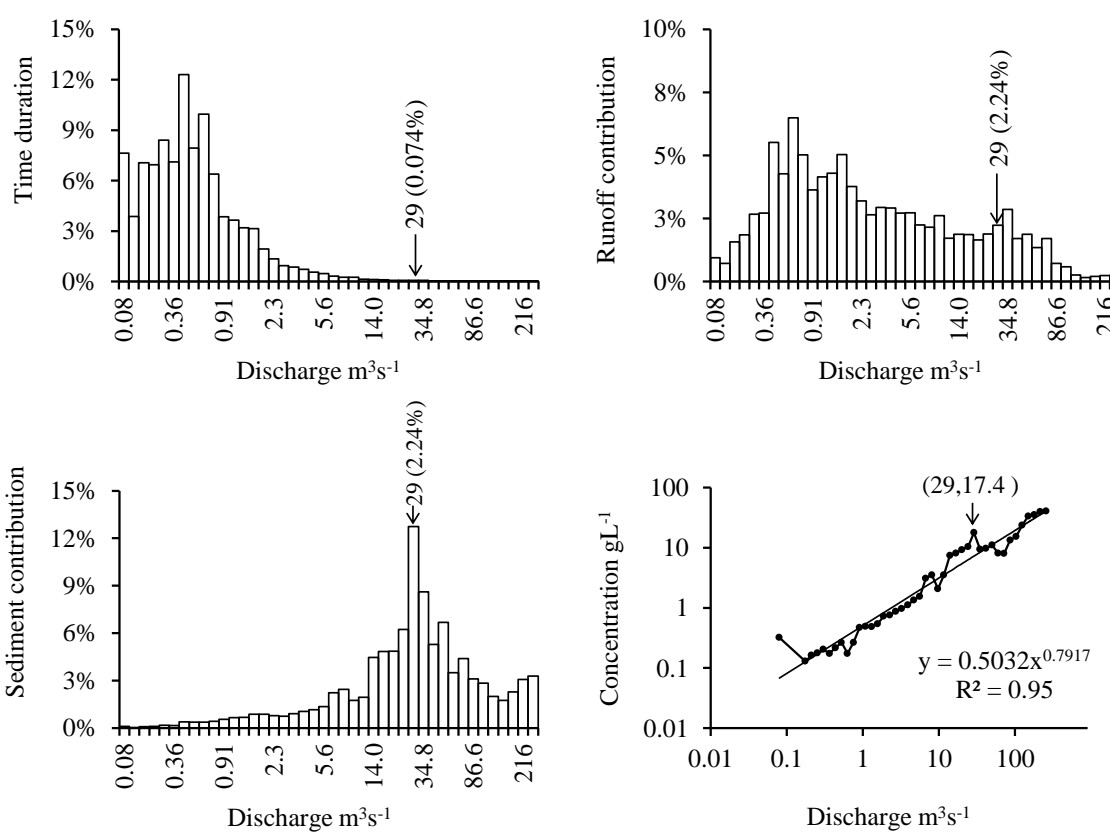

**Figure 6: Duration, water and sediment input per class, and sediment rating curves, for the subdivision into classes of geometric progression of common ratio 1.2**



The determination coefficient and Nash-Sutcliffe coefficient (1970) of the rating curve, $C^k = a \, Q^{k \, b}$, are satisfactory for the three types of subdivisions used in this study (Table 1). However, the best performances are obtained with the subdivision in geometric progression which also allows a better quantification of the sediment supply (Table 1). A more detailed and comparative analysis of the results obtained by using the subdivision into classes of equal amplitude 1 m³ s⁻¹ and the subdivision

into geometric progression intervals of common ratio 1.2 is provided hereafter.

## 4.2 Analytical and statistical approach

The analytical approach requires to build a probability density function f(Q) representing the distribution of flow frequencies as well as a curve g(Q) representing the solid discharge $Q_S$ as a function of the water flow Q. The study shows that these two curves are closely related to the types of subdivisions used. For the subdivision into classes of equal amplitude 1 m³ s⁻¹ and the

one with geometric progression of common ratio 1.2, the adjustment of flow frequency distribution to the log-normal and log-Gumbel probability distributions are satisfactory (Fig. 7) with a slight advantage for the log-Gumbel distribution. The highest difference for a class between the empirical and theoretical (log-Gumbel) frequency distributions was 4.1% for the subdivision into classes of equal amplitudes and 4.8% for subdivision into geometric progression. Characteristic parameters associated with the subdivision into classes of equal amplitudes and the one into geometric progression are ($\mu$ = -0.4148, $\delta$ = 0.6572) and

($\mu$ = -0.7180, $\delta$ = 0.9649), respectively. However, the dominant discharges obtained when then log-Gumbel distribution is considered are very low: $Q_D = 0.64$ m³ s⁻¹ for the subdivision into equal classes of amplitude 1 m³ s⁻¹ (with b = 0.8031), and $Q_D = 0.62$ m³ s⁻¹ for the subdivision into geometric progression (with b = 0.7917). The use of a lognormal distribution leads to slightly higher values for $Q_D$: 0.92 m³ s⁻¹ for the subdivision into classes of 1 m³ s⁻¹ and 1.02 m³ s⁻¹ for the subdivision into geometric progression, far from the dominant discharges obtained from the histograms, 29.5 and 29.01 m³ s⁻¹ (Table 1).

The annual maximum flow rate series, $Q_{MAX}$, and the $Q_{Y1/2}$ annual half-load discharge series fit log-normal distributions (Fig. 8). These two probability distributions make it possible to evaluate the recurrence interval of $Q_D$ values. The two subdivisions (equal classes of amplitudes 1 m³ s⁻¹ or in geometric progression of common ratio 1.2), with very close $Q_D$ values, give similar recurrence intervals: the return period is 2.2 years for the annual series $Q_{MAX}$ and 6.9 to 7 years for the annual series $Q_{Y1/2}$ (Table 2). The difference of nearly five years between these two estimates clearly illustrates the importance of the choice of

the annual series considered to estimate the recurrence interval of the dominant discharge. Their meaning is different and can be explained: they indicate that the effective discharge is observed at least once in a hydrological year roughly every two years at the gauging station, but that half of the yearly sediment supply is carried by flows higher than the effective discharge only every 7 hydrological years.




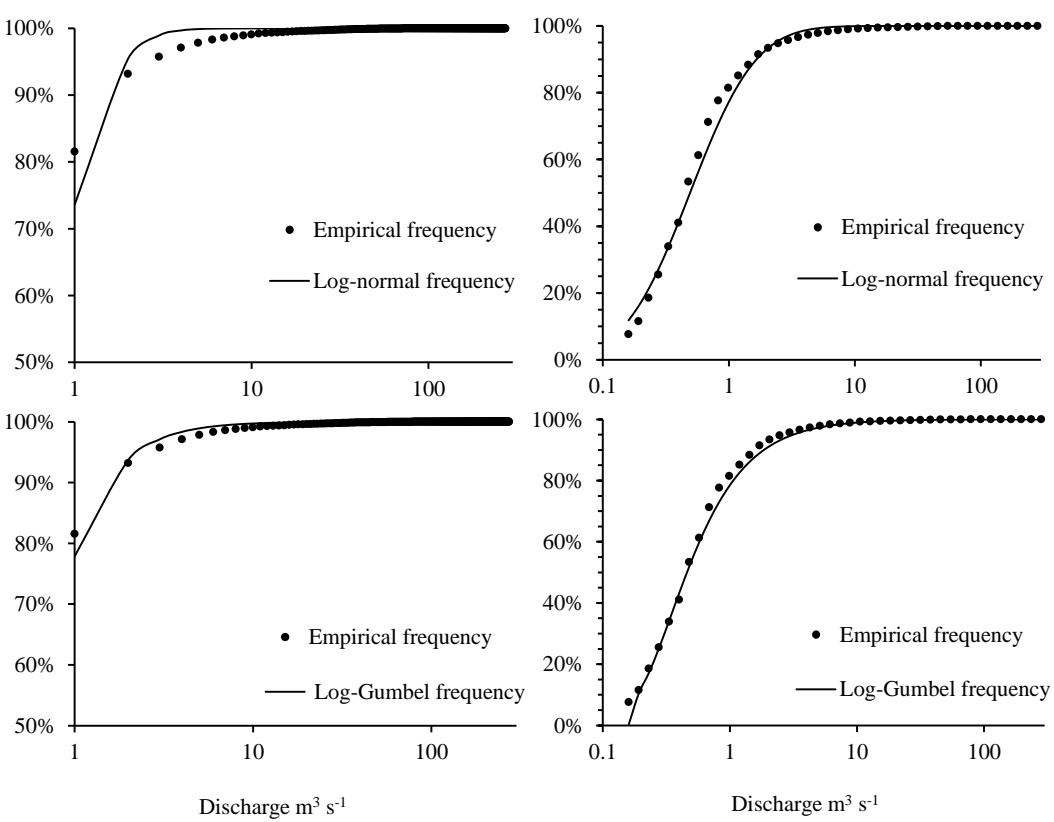

**Figure 7: Adjustment of the frequency distribution of flows to the log-Gumbel probability distribution: on the left, according to a subdivision into equal classes of amplitudes 1 m³ s⁻¹, and to the right, according to a subdivision into geometric progression of common ratio 1.2.**

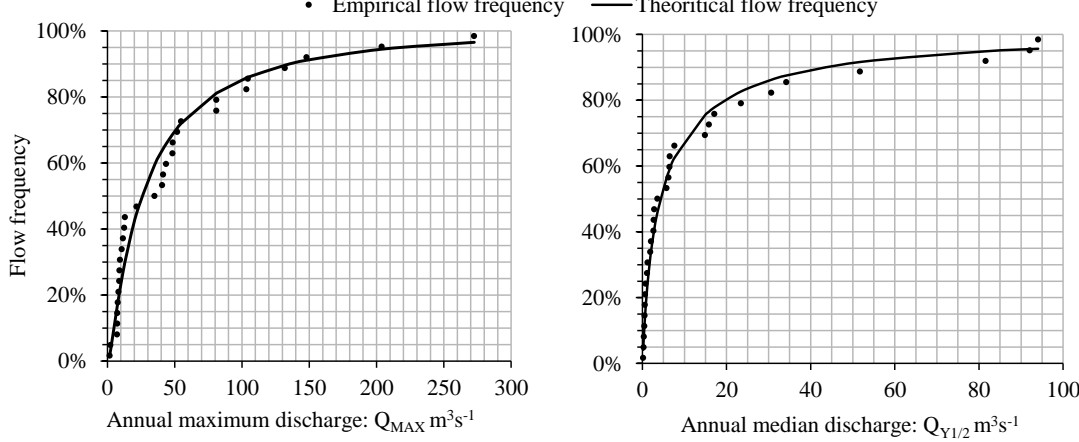

**Figure 8: Adjustment to the log-normal distribution of maximum annual discharges, $Q_{MAX}$, and median annual discharges, $Q_{Y1/2}$.**





**Table 2: Recurrence intervals, R.I. $Q_{MAX}$ and R.I. $Q_{Y1/2}$, of the dominant discharge $Q_D$ calculated for the subdivisions into classes of equal amplitude 1 m$^3$ s$^{-1}$ and of geometric progression.**

| Method for $Q_D$ calculation | $Q_D$ (m$^3$ s$^{-1}$) | R.I. $Q_{MAX}$ (year) | R.I. $Q_{Y1/2}$ (year) |
|---|---|---|---|
| subdivision into classes of equal length 1 m$^3$ s$^{-1}$ | 29.50 | 2.18 | 7.02 |
| subdivision in geometric progression (1.2) | 29.01 | 2.16 | 6.91 |

## 4.3 Characterization of flows and associated sediment loads

The evolution of water and sediment supplies as a function of increasing flows distributed in classes with geometric progression
is non-synchronous (Fig. 9). The evolution of the sediment supply as a function of the water supply shows varying behaviours
identified by branches in Fig. 9-c. The 1$^{st}$ branch, located between points (1) and (2), ends with the modal class [0.57; 0.69 m$^3$
s$^{-1}$[. These frequent flows last on average 71.6% of the time and contribute up to 26.8% in water supply and 1.7% in sediment
supply. Sediment supplies and concentrations remain low and almost stable with increasing discharges, less than 0.4% for
sediment inputs and 0.2 g L$^{-1}$ for concentrations. The second branch (2)-(3) consists of classes contained in the interval [0.69;
12.7 m$^3$ s$^{-1}$[. Over this range, the sediment supply and concentrations are increasing moderately while the discharge is clearly
increasing. Flows last on average 28.1% of the time and contribute to 51.8% of water supply and 18.4% of sediment supply.
Concentrations range from 0.4 to 3.5 g L$^{-1}$, the average being 1.1 g L$^{-1}$. The third branch (3)-(4) consists of 5 classes with
adjacent contributions in water, between 1.64 and 2.22%, with highly increasing sediment concentration and supply with
increasing discharge. They contribute to 0.46% of the time, 9.5% of water supply and 33.1% of sediment supply. The average
concentration per class varies between 3.5 and 17.8 g L$^{-1}$, the average being worth 10.9 g L$^{-1}$. The highest efficiency is attributed
to class (4) with effective flow [26.4; 31.7 m$^3$ s$^{-1}$[. The fourth branch, consisting of the only class [31.7; 38.0 m$^3$ s$^{-1}$[, is
distinguished by a dilution of water supplies. Compared to the efficient flow class, which precedes the 4$^{th}$ branch, the volume
of water produced increases from 2.24 to 2.67% and the concentration decreases from 17.8 to 9.4 g L$^{-1}$. The frequency of the
4$^{th}$ branch flows, 0.079%, is similar to that of the dominant class (0.074%). Discharges higher than 38.0 m$^3$ s$^{-1}$ are the last
branch where water and sediment supplies are synchronously decreasing, while concentration is increasing. Such flows are
rare but effective in terms of sediment production.





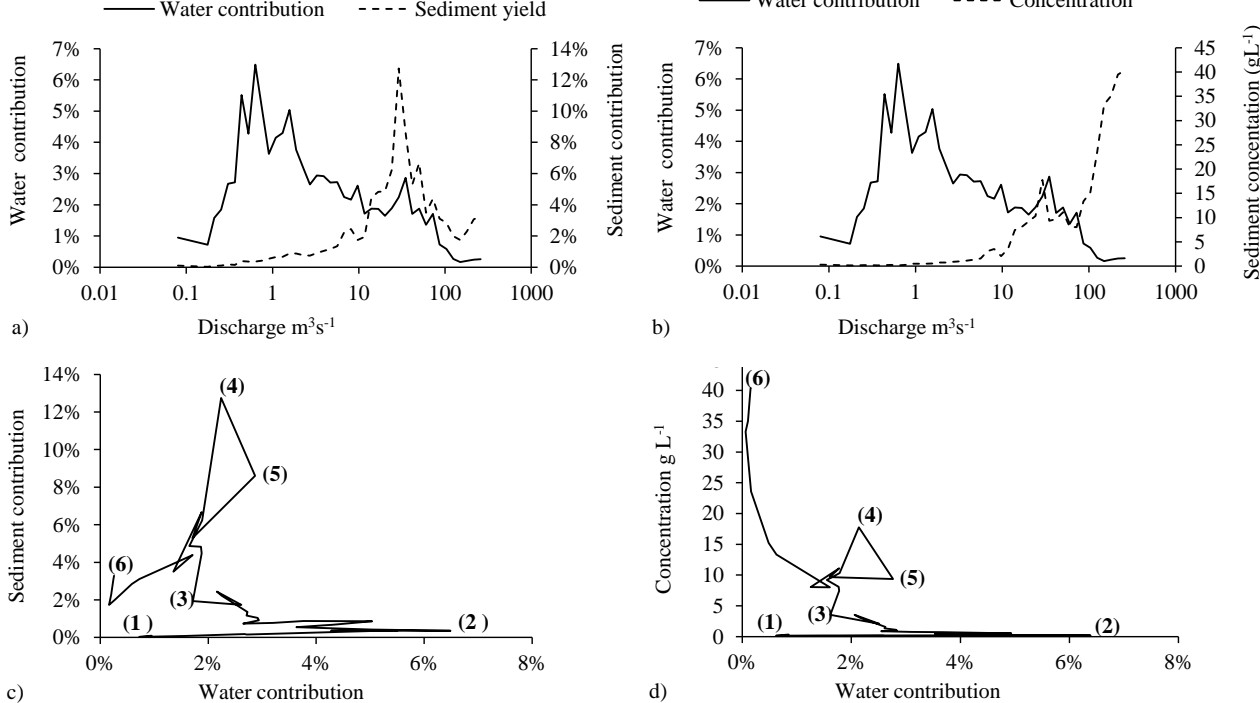

Figure 9: Water and sediment supplies, and sediment concentrations, with the subdivision into classes of geometric progression.

## 5 Discussion

### 5.1 Pre-processing of data of the gauging station

Half-hour sampling carried out by the ANRH is unsuitable during the Wadi Sebdou flash floods, which produce more than 80% of the total sediment load in 1% of the time, with an estimated average concentration of 10.3 g L$^{-1}$. To overcome the presence of empty classes, Biedenharn et al. (2001), Goodwin (2004) and Crowder and Knapp (2005) propose to downgrade, subjectively, the number of classes by readjusting their amplitude to cover all classes in information. Another alternative applied in this study is to refine the dataset, by interpolation between measurements. The refinement tested in this study has

the advantage of not modifying the water and sediment budgets brought by the Wadi compared to the original series, since the interpolation is linear. The discharge step chosen for the interpolation, close to the low flow at the hydrographic station, also makes it possible to cover all classes of the different considered subdivisions and thus to make it possible to calculate the effective discharge. Note that a similar method has already been applied by Biedenharn et al. (2001) and Gray et al. (2015) to refined data from monthly step to daily steps, and by Simon et al. (2004) and Lenzi et al. (2006) from daily and hourly

measurements to a finer time step of 15 or even 5 minutes. In other studies, for which the frame of reference is the daily time step, instantaneous measurements are replaced by daily averages (Andrews, 1980, Nolan et al., 1987, Emmett and Wolman, 2001).



## 5.2 Methodology to identify the dominant class

### 5.2.1 Choice of a subdivision of discharges

The quality of graphs and the error on water and sediment supplies made it possible to compare subdivisions and select those that are able to represent the flows and to identify the effective discharge. Several studies dedicated to dominant discharge
class focused exclusively on the graphical aspect by readjusting the interval amplitude with equal classes until a dominant class appears outside the first and last classes (Benson and Thomas, 1966; Pickup and Warner, 1976; Andrews, 1980; Hey, 1997; Lenzi et al., 2006; Roy and Sinha, 2014). However, this approach remains subjective (Sichingabula, 1999; Biedenharn et al., 2001; Goodwin, 2004) and poses a dilemma. Reducing the class amplitude can make the dominant class emerge outside the two extreme classes, but this can bring up empty classes which, conversely, require increasing the amplitude for each class
to be covered. Where appropriate, the series is considered non-compliant with the selection criteria and does not allow to identify the dominant class (Crowder and Knapp, 2005). To avoid such situations, Bienderhan et al. (2000) recommended the use of adequately provided datasets covering at least 10 years of measurements.

Yevjevich's (1972) proposal, based on statistical concepts, to use between 10 and 25 classes of amplitude less than $s/4$, where $s$ is the standard deviation of the flow series, is difficult to apply to the Wadi Sebdou. The standard deviation, which can be
calculated from $s^2 = \frac{1}{T}\sum_i \Delta t_i (Q_i - \bar{Q})^2$ where $\bar{Q} = \frac{1}{T}\sum_i \Delta t_i Q_i$ and $\Delta t_i$ is the elementary time interval (Eq. 1), gives $s/4 = 0.77$ $m^3 \, s^{-1}$ for the Wadi Sebdou. Subdivision into classes of equal $s/4$ amplitudes would require 355 classes to cover the range of flows. In a stream with such high flow variability, the strong flow asymmetry has a negative impact on the representativeness of flows and sediment discharges, especially for low flow classes that cover most of the water supply. This suggestion does not seem appropriate for wadis.
In this study, two types of subdivisions other than the classical subdivision with classes of equal amplitude were examined: discharge classes corresponding to equal water supply, and a geometric progression of flows. The subdivisions into classes of equal amplitude 1 $m^3 \, s^{-1}$ and the subdivision into classes with geometric progression best represented liquid and sediment supplies (Table 1) and are used to characterize the Wadi Sebdou flows. They give dominant discharges ($Q_D = 29.5 \, m^3 \, s^{-1}$ and $Q_D = 29.0 \, m^3 \, s^{-1}$, respectively) very close to each other and to the half-load discharge $Q_{Y1/2} = 29.8 \, m^3 \, s^{-1}$. This result is in
perfect agreement with Vogel et al. (2003). The half-load discharge, which is simple to compute, is used by several authors (Doyle and Shields, 2008; Klonsky and Vogel, 2011; Ferro and Porto, 2012; Gray et al., 2015) and has been generalized to identify the dominant discharge conveying various solid or dissolved matters (nutrients, sand, accidental pollution, etc.), especially for the study of ecological aspects and environmental management (Vogel et al., 2003, Doyle et al., 2005, Wheatcroft et al., 2010).

### 5.2.2 Influence of the rating curve $Q_s = g(Q)$

The sediment supply calculated from data ($t_i$, $Q_i$, $C_i$) (reminder: the supply is the same with the 6,947 initial values as with the 40,081 values, linearly interpolated) provided a reference to evaluate the ability of a rating curve to estimate sediment




discharges from water flows. This rating curve $g(Q^k) = Q_s{}^k = a\,Q^{k\,(b+1)}$ established from the series $(Q^k, C^k)$ generates errors we call hereafter 'of the first type', which we must specify. The sediment supply associated to the $k^{th}$ discharge class is as follows:

$$\Delta Y^k = \frac{1}{Y_T} g(Q^k)\Delta T^K \tag{28}$$

where $Q^k$ and $\Delta T^K$ are the center and the duration of flows corresponding to a given class.

On the Wadi Sebdou, despite a correct estimate of the total sediment supply for the two subdivisions of equal amplitude 1 m$^3$ s$^{-1}$ and in geometric progression (Figure 10, Table 1), the rating curve $Q_s{}^k = a\,Q^{k\,(b+1)}$ generates errors that induce a shift in the class of dominant discharge (Figure 11). These errors result from the construction of the calibration curve which is based on a minimization of differences between observed and estimated sediment discharges on a logarithmic scale. The subdivision into classes of equal amplitude leads to a value of effective discharge, $Q_D=1.5$ m$^3$ s$^{-1}$, very low in comparison with the one calculated

from initial data using eq. 5 (29.5 m$^3$ s$^{-1}$). The use of a rating curve for $Q_s$ with the subdivision in geometric progression results in an effective discharge of 72.2 m$^3$ s$^{-1}$, well above the value obtained directly (29.01 m$^3$ s$^{-1}$). These offsets are explained because the actual sediment discharges associated with each class are around the rating curve $g(Q_k)$, sometimes below or sometimes above (Figs 3 and 6). For both subdivisions, the empirical average sediment concentration observed in the dominant class is well above the rating curve. As a result, the rating curve greatly underestimates the sediment supply in this class.

Combined with the flow frequency, the supply is lowered compared to other classes where the model overestimates the average concentration.

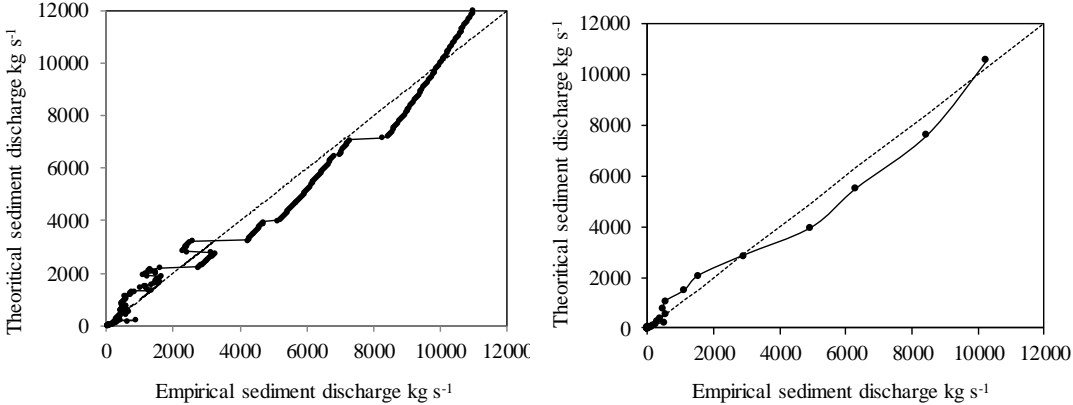

**Figure 10: Comparison between the analytical sediment supply by class given from the rating curve (power model) and the elementary contributions $Q_{S.Obs} = \frac{\Delta Y^K}{\Delta T^K}$ where $\Delta Y^K$ and $\Delta T^K$ are obtained from Eq. 3: for a subdivision into classes of equal**

**amplitudes 1 m$^3$ s$^{-1}$ (left) and with a geometric progression of common ratio 1.2 (right).**





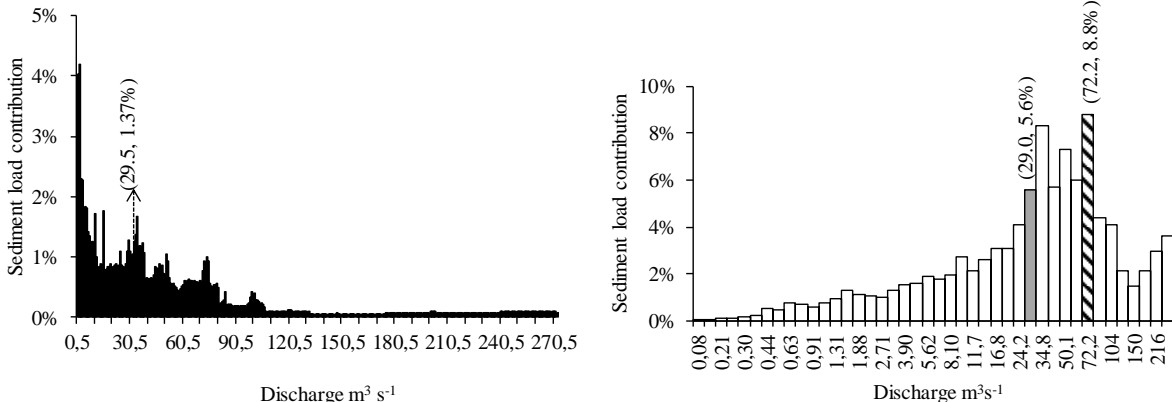

**Figure 11: Sediment load histogram established using the sediment rating curve: for a subdivision into classes of equal amplitudes 1 m³ s⁻¹ (left) and with a geometric progression of common ratio 1.2 (right).**

### 5.2.3 On the use of a rating curve for $Q_s$ and a probability distribution for Q

When the flow frequency is represented by a probability distribution, the sediment load histogram can be built from the rating curve and this distribution. However, it should be remembered that for a continuous random variable such as water discharge, the theoretical probability at a point does not exist in the probabilistic sense, but necessarily refers to an interval. Thus, the contribution of a class, $I_k$, can be quantified by:

$$\Delta Y^k = \frac{1}{Y_T} g(Q^k) \left[ \int_{I_k} f(Q) dQ \right] T \tag{29}$$

where $Q^k$ is the center of the $I_k$ interval, f is the probability density function, T is the total duration of the study period, and $Y_T$ is the total sediment yield. On the Wadi Sebdou, the rating curve and the two log-normal and log-Gumbel probability distributions correctly represent the water and sediment supplies (Table 1, Fig. 7). The analysis of errors associated to the subdivision in geometric progression and the log-normal distribution (Figure 12) shows that above 18.3 m³ s⁻¹, the ratio actual frequency on analytical frequency is very high and varies from 11.6 to more than $10^5$ for the log-normal distribution.

Consequently, the analytic supply is minimal compared to the load calculated from elementary contributions for these flows, and the total analytical supply given by Eq. 29 underestimates by 79% the sediment supply established by Eq. 5. With the log-Gumbel distribution, the ratio varies between 0.16 and 1.69 and the total analytical yield overestimates by 35% the one deduced from elementary contributions (Eq. 5). The offset is also high when flows are subdivided into equal classes of amplitude 1 m³ s⁻¹. Compared with the total analytical yield, low flow rates seem to be the most effective. The class [1, 2 m³ s⁻¹[ dominates

with a contribution of about 6% for the log-normal distribution and 15.7% for the log-Gumbel distribution. In this case, the total sediment load estimated from the log-normal and log-Gumbel distributions underestimates by 84% and 66%, respectively, the empirical load.





This example shows that the product of the theoretical frequency distribution generates errors of a second type which are not taken into account in the construction of the sediment load histogram which, as a result, poorly represents the distribution of the sediment supply (Fig. 13).

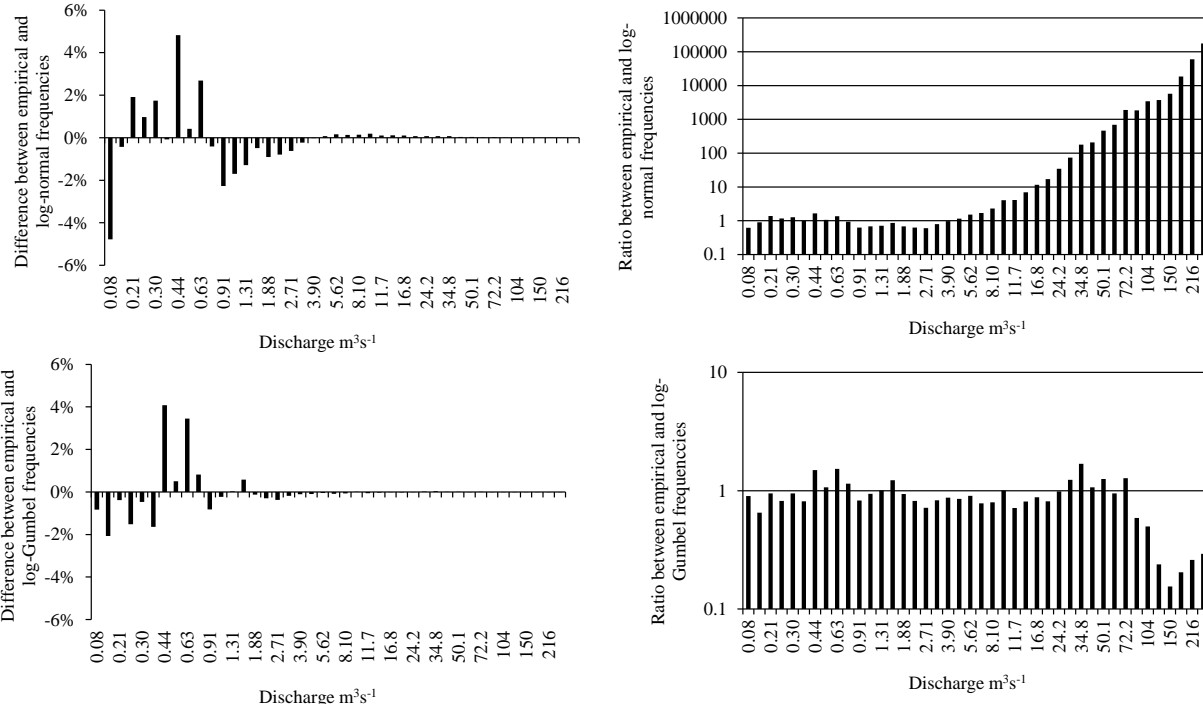

5 **Figure 12: Analysis of errors (difference and ratio) between observed and theoretical frequencies: for the log-normal distribution (above) and for the log-Gumbel distribution (below).**

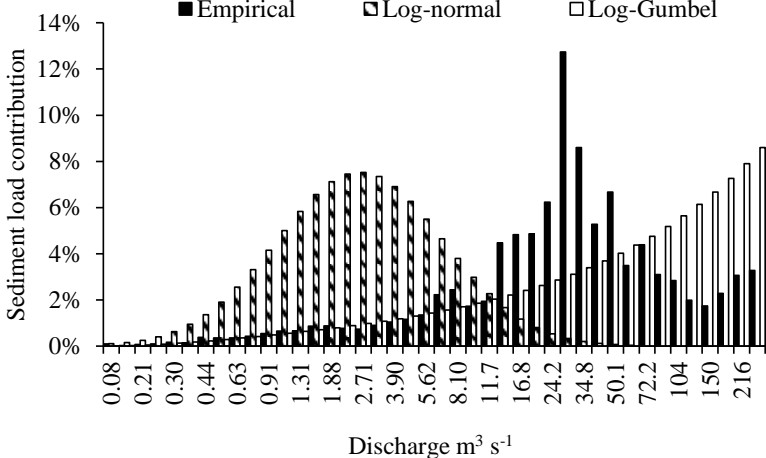

**Figure 13: Three sediment load histograms obtained from the dataset, from the product of the rating curve times to the log-normal distribution of discharges, and from the product of the rating curve times to the log-Gumbel distribution.**



Since the function f increases until the mode, $Q_{mode}$, where it reaches its maximum, the dominant discharge $Q_D$ is greater than $Q_{mode}$ by construction (Fig. 1). The difference between $Q_D$ and $Q_{mode}$ depends on the growth of the function g and the decrease of the function f. However, for wadis, the scarcity of flood events and the dominance of low flows (80% of flows are less than 1 $m^3$ $s^{-1}$ on the Wadi Sebdou) require the use of a probability density function with a pronounced dissymmetry where, after the

mode, the decay is rapid. In this context, only the log-normal and log-Gumbel distributions have shown a satisfactory fit to subdivisions in classes of equal amplitude 1 $m^3$ $s^{-1}$ and in geometric progression (Fig. 7).

The analytical solution applied to the data gives a very weak dominant discharge in comparison with the one calculated using the formula of Nash (1994) on the Wadi Sebdou. It should be noted that, due to the neglect of the Q term in the denominator of the density function of the log-normal distribution (error already underlined by Vogel et al., 2013), the analytical solution

proposed by Nash (1994) induces excessive effective discharge. The analytical expressions giving $Q_{Mode}$ and $Q_D$ (Eq.21, 22, 26 and 27) partly explain the low value of the dominant discharge found for the Wadi Sebdou which is mainly dependent on the low value of the $\mu$ parameter due to the specific hydrologic regime in semi-arid environments, where the annual modulus is very low, often below 1 $m^3$ $s^{-1}$. The dominant discharge is thus very close to the mode. As a result, the analytical approach seems to be suitable only for rivers where extreme flows are less distant from the mode than on wadis.

Another point deserves a remark in the calculation of the effective discharge from f and g, for the general case. Whatever the probability density function f, the sediment transport efficiency curve is given by $h(Q) = a\ Q^{(b+1)}\ f(Q)$. Thus, the effective discharge, solution of the derived function $h'(Q) = 0$, is independent of the parameter *a* and depends only on the parameter *b*. In other words, the dominant discharge depends exclusively on characteristics of the watercourse since parameter *b* is commonly considered as an indicator of the erosive power of the watercourse (Leopold and Maddock, 1953; Roehl, 1962;

Fleming, 1969; Gregory and Walling, 1973; Robinson, 1977; Sarma, 1986; Reid and Frostick, 1987; Iadanza and Napolitano, 2006; Yang et al., 2007). However, the suspended sediment load in rivers is strongly influenced by characteristics of the basin as well, where slopes contribution to sediment supply is high (Megnounif et al., 2013), and even sometimes higher than the one of the hydrographic network (Roehl, 1962; Gregory and Walling, 1973; Duysings, 1985; Asselman, 1999).

Finally, it should be noted that introducing a density function necessarily gives a monomodal sediment transport efficiency

curve, whereas this is not necessarily the case. Pickup and Warner (1976), Carling (1988), Phillips (2002), Lenzi et al. (2006) and Ma et al. (2010) reported the existence on some sites of a bimodal dominant flow. Hudson and Mossa (1997) pointed out that sediment load histograms present a variety of forms, including bimodal and complex forms, that differ from the unimodal form identified by Wolman and Miller (1960). In addition to the monomodal sediment load histograms, Ashmore and Day (1988) distinguished three other kinds of histograms: bimodal, multimodal and complex. Of the 55 basins studied by Nash

(1994), 29 are bimodal and 9 are multimodal.

## 5.3 Recurrence interval of the effective discharge

The recurrence interval of a given discharge is necessarily deduced from a probability distribution adjusted to a series of annual flows. This is the case for the effective discharge, often estimated from the annual maximum flow series (Biedenharn et al.,



2001; Crowder and Knapp, 2005; Lenzi et al., 2006; Ferro and Porto, 2012; Gao and Josefson, 2012; Bunte et al., 2014). However, the dominant discharge is, by definition, a moderate flow with a high probability of occurrence (Wolman and Miller 1960). This study has also verified that the dominant discharge is comparable to the half load discharge of Vogel et al. (2003). As a result, the derivation of the dominant discharge from the statistical population of maximum annual discharge series is not straightforward.

This assumption, verified on the Wadi Sebdou, led us to propose an estimate of the recurrence interval from a series of central values, such as the series of half load discharge $Q_{Y1/2}$. Consequently, the dominant discharge will be among the low values of the annual maximum discharge series and will necessarily provide a shorter recurrence interval than that deduced from the $Q_{Y1/2}$ quantile series. For the Wadi Sebdou, the difference between these two recurrence intervals is around 4.8 years (Table 2).

## 5.4 Sensitivity of the dominant class to the environment

Biedenharn et al. (2001) suggest to carefully study long (over 30 years) data series (liquid flow, sediment concentration, and flow frequency) and to ensure that the hydrological regime of the watershed did not undergo a significant change in flow rates or sediment production in the long term. Change can be attributed to climate change (Zhang and Nearing, 2005; Ziadat and Taimeh, 2013; Liu et al., 2014; Achite and Ouillon 2016) or anthropogenic actions (Cerdà, 1998a, 1998b; Liu et al., 2014), such as intensification of agriculture (Montgomery, 2007; Lieskovský and Kenderessy, 2013), deforestation (Walling, 2006), forest fires (González-Pelayo et al., 2006; Cerdà et al. 2010) or urbanization (Graham et al., 2007; Whitney et al., 2015). In the study area, and like North Africa and the Maghreb, there has been a continuous drought since the mid-1970s (Giorgi and Lionello, 2008; Achite and Ouillon, 2016; Zeroual et al., 2016). Overall, decreasing rainfall is more concentrated over time (Ghenim and Megnounif, 2016), which increases the susceptibility of soils to erosion (Shakesby et al., 2002; Achite and Ouillon, 2007; Bates et al., 2008; Vachtman et al., 2012). Megnounif and Ghenim (2016) showed that sediment production, which is strongly correlated with irregular rainfall (Achite and Ouillon, 2007), increased significantly in the late 1980s, with a pivot in 1988. After 1988, the annual sediment yield was on average 7 times higher compared to the previous period (Megnounif and Ghenim, 2013).

The application of a subdivision of discharge classes into geometric progression at the Wadi Sebdou for the two periods 1973-1988 and 1988-2004 confirmed the change in the watershed functioning, with a bimodal sediment supply distribution for the first period (Fig. 14). For 1973-1988, the class [6.1; 7.4 m³ s⁻¹[, which includes the effective discharge $Q_D = 6.7$ m³ s⁻¹, contributes 7.5% of the total sediment yield. These relatively frequent flows last on average 0.5% of the annual time, i.e. 1.83 days, which represents 6.3% of the annual duration of floods which, for this period, last 7.65% of the annual time. The second peak is observed for Q = 34.8 m³ s⁻¹ representing the class [31.7; 38.0 m³ s⁻¹[. Flows of this class are rare and last only 0.09% of the annual time (7h45), but carry 5.9% of the total sediment yield. Over the period 1988-2004, the distribution of sediment supply became essentially monomodal (Fig. 14) with a dominant flow $Q_D = 29.0$ m³ s⁻¹. During this second period, an important





sediment contribution was also observed at high discharges (> 110 m³ s⁻¹, up to 273 m³ s⁻¹) that were not sampled during the period 1973-1988.

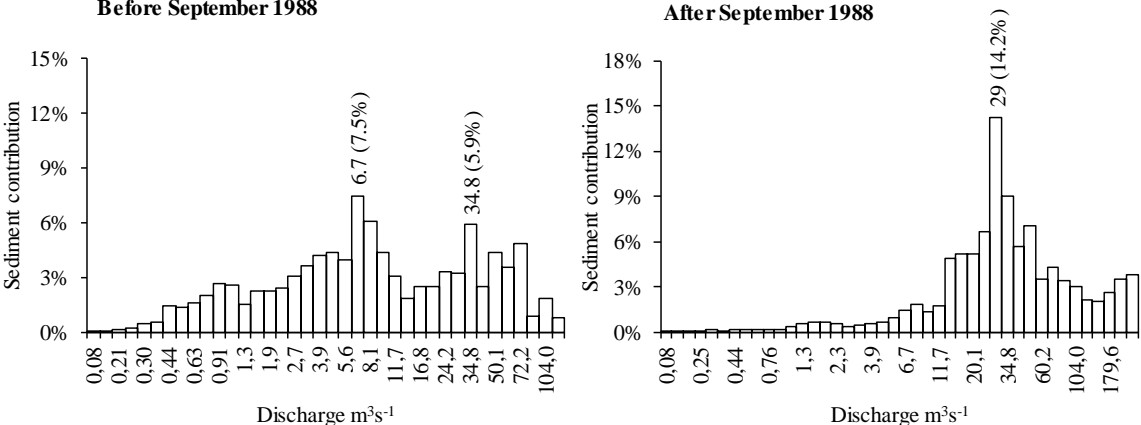

**Figure 14: Sediment supply by class for the subdivision in geometric progression: for 1973-1988 (left) and for 1988-2004 (right).**

## 5.5 Simple ways to estimate the effective discharge

The dominant discharge, as we have seen, corresponds to a discharge class for which the sediment contribution increases considerably with Q. There is a simple way to estimate it from the cumulative sediment yield curve assigned to ordinal discharges. On the Wadi Sebdou (Fig. 2), this curve clearly shows two flow classes for which sediment accumulation increases the most – those for which this curve has a maximum derivative –, around 15 m³ s⁻¹ and 29-30 m³ s⁻¹. The value of $Q_{Y1/2}$ is also easy to estimate from the cumulative sediment yield curve, since it corresponds to the discharge corresponding to 50% of the inputs (i.e. about 30 m³ s⁻¹ in Fig. 2 for the whole period). A more detailed analysis like the one presented in this paper will then make it possible to determine with more precision the class, the dominant discharge and its return period.

Alternatively, the dominant class is also distinguished on the curve showing average concentrations by classes as a function of discharges (Fig. 3, Fig. 5). For every selected subdivision, the dominant class appears as the one whose highest point is above the rating curve: at 29.5 m³ s⁻¹ for classes of length 1 m³ s⁻¹ (Fig. 3); at 29 m³ s⁻¹ for classes in geometric progression (Fig. 6).

## 6 Conclusion

From a time series of flow and concentration data, a direct calculation provides estimates of water and sediment supplies by summing the elementary contributions. This gives access to seasonal or annual values, and to the analysis of their variability. Sediment dynamics can also be analyzed from discharge and sediment yield histograms by water discharge classes. On the Wadi Sebdou, we have shown that an appropriate choice of subdivisions makes it possible to minimize the difference between the flows estimated and measured at less than 10% ($\tau_Y = \tau_R = 8.8\%$ for classes of equal amplitude 1 m³ s⁻¹, Table 1) or even





less than 1% ($\tau_Y = \tau_R = 0.3\%$ for classes in geometric progression of common ratio 1.2, Table 1). Classes thus defined make it possible to determine a dominant class in the sense of sediment yield $Q_D$ (29.5 or 29 m³ s⁻¹ according to the two classifications mentioned above) which is similar to the median flow in the sense of sediment yield $Q_{Y1/2}$ (29.8 m³ s⁻¹) on the Wadi Sebdou. Other classifications have proved to be able to estimate the effective discharge (with a lesser precision) but unable to provide

good estimates of the water and sediment supplies (classes of equal amplitude greater than 1 m³ s⁻¹), or able to estimate these supplies but unable to estimate the effective discharge (classes with equal water supplies).

In order to determine an analytical solution, the introduction of a rating curve between the $Q^k$ and $C^k$ series considered to build the histogram induced an additional bias with respect to the direct calculation for the sedimentary yield. On the Wadi Sebdou, this bias, which depends on the choice of subdivisions, can be reduced by 6 to 7% as indicated by $\tau_{MY}$ (Table 1), which is

acceptable with regard to either the uncertainties of measurements, or sometimes of inappropriate or insufficient sampling (Coynel et al., 2004). Previous work (Megnounif et al., 2013) has shown the importance of hysteresis phenomena on this basin which induces a strong dispersion of instant parameter pairs (Q, C) and a bias in the estimation of supplies using a rating curve. The rating curve based on average values by flow class, of defined length in geometric progression, appreciably improves statistical performances in the computation of the sediment supply (higher value of $R^2$, 0.95, and better Nash-Sutcliffe criterion,

0.93, compared to any other method – see Table 1). It will be interesting in future works to analyze whether this is a singular phenomenon or whether the use of a rating curve based on subdivision classes rather than on instantaneous measurements makes it possible to improve the calculation of sediment yield compared with more conventional methods.

The coupled use of a calibration curve and a probability distribution of discharges should make it possible to obtain an analytical solution for the dominant discharge. On the Wadi Sebdou, the log-normal and log-Gumbel distributions were most

likely to reproduce the observed regime, characterized by a very weak mode and the existence of flash floods. The log-Gumbel distribution was better than the log-normal distribution for predicting frequencies and therefore return times, especially at flow rates <72 m³ s⁻¹ (Fig. 12). On the other hand, they are still not precise enough to be used to estimate sediment yield during flash floods. Indeed, the estimation of sediment supply is extremely sensitive to two stages: the rating curve $Q_s$-Q and the estimation of the flow frequency. An error on the first step (error of the first type) and an error on the second (error of the

second type) multiply and have no reason to compensate each other.

This study made it possible to compare the validity of different methods to estimate the dominant discharge and its return period on a wadi, in a semi-arid environment. Two return periods were identified for the effective discharge: one for the interval between two hydrological years with occurrence of the effective discharge (from the annual maximum flow rate series, $Q_{MAX}$), and one for the duration between two hydrologic years for which half of the sediment supply at least is carried by flows higher

than the effective discharge (from the $Q_{Y1/2}$ annual half-load discharge series).

Flows of the dominant class carry the most sediment in the watercourse. It should be possible to link them to major processes of erosion, transport and deposition that occur in the watershed. Lenzi et al. (2006), who have observed a bimodal sediment contribution for a mountain river in the Alps in Italy, attributed the first modal class, of low but more frequent flow, to the shaping of channel, and suggested that the second-class flows, larger in magnitude but of low occurrence, would be responsible





for the macroscale shape of the watercourse. On the Wadi Sebdou, we also observed a bimodal distribution. However, it is difficult to conclude because the secondary mode of distribution obtained for the first period (1973-1988) became the dominant mode of distribution later (1988-2004). The modes move with the hydrological regime towards higher and higher sediment yields, in line with what has been observed on most of watersheds studied over the last decades in the semi-arid environments

of Algeria (e.g. Achite and Ouillon, 2016). Applying this method to other watersheds will undoubtedly allow us to go further in the analysis of dominant discharges and in their dynamics, in a context of global change.

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
