# Peer review of "Mean and analytical methods to characterize the efficiency of floods to move sediment in a small semi-arid basin"

_Hydrology and Earth System Sciences, 2018_

## Referee Comment (RC1) · Anonymous Referee #1 · 4 Jun 2018

The paper is interesting and well within the scope of the journal, nevertheless it needs to be reinforced with reference to several points, mainly regarding paper clarity and organization. The title should more informative and let the reader know that it deals with sediment transport and hydrological implications. The introduction should be clearer with respect to the paper's objectives and novelty. While it is clearly stated that three methods for individuation of discharge classes are compared, it is not evident how many and which methods are compared with reference to the evaluation of effective discharge, and other related methods including evaluation of return times. While the

paper is strongly focused on issue related to the representativeness of available measures (discharge and concentrations), only few explanations are provided about physical processes related to sediment transport. The eventual presence of hysteresis which is at the core of many works on the topic is here barely mentioned. A figure representing the basin and its position is missing and basin description is almost entirely addressed to other papers referenced. Also, the organization of section 3 Methodology, does not help the reader in understanding what the authors are mainly presenting and comparing. Making an exception for subsections 3.2 about class intervals, all other parameters are presented without any specific order of hierarchy. Subsection 3.3 about hydrometric measurements and pre-processing could be probably moved in section 2. The evaluation of the Half-load discharge does not actually add any knowledge insight apart for a weak literature comparison. The same applies to the subsection 3.7 Recurrence interval which is a mere evaluation of a certain discharge compared to the distribution of maximum and mean discharge. Figure 3, upper portion, should be placed in a different scale, data are almost invisible. Last sentence of page 17 is not clear, should be rephrased. The striking difference between analytical and statistical approaches is simply distressing and does not find a satisfactory justification. Figure 9c and 9d, are hard to understand. Maybe that placing indications of number (1) to (6) on the time series (i.e. in figures 9a and 9b) may help. The discussion section is quite long and not always add useful information. Some parts could be shortened and moved to other subsections, for the sake of paper structure and readability. Other parts may be even canceled like lines 1-6 at page 25 or lines 1-10 page 26 or the entire 5.5 subsection.

---

## Referee Comment (RC2) · Anonymous Referee #2 · 26 Jul 2018

I think this is an interesting manuscript potentially reporting on an interesting data set and a thorough analysis. However, in my opinion the authors fail to make a convincing case for while this paper is a significant and original contribution to the scientific literature. The introduction is mostly a methodological introduction. There is some text in section 3.3 which does try to describe the scientific context of this particular study which could perhaps be used as a starting point for a more focused introduction, reviewing the literature and identifying knowledge gaps.

Also, the manuscript is quite long as there is a lot of fundamental methodology in-

cluded. I think it would be more readable if the focus was more on the original aspects of the analysis with less reference to standard methods used.

The conclusion is very long. I would suggest a more concise set of conclusions would help to communicate the potential importance of the paper to readers.

In summary: this is potentially an interesting paper, but there is much that can be done in order to improve the quality of the presentation.

Other comments

Section 3: Discharge is Q, concentration is C, and the product of the two is Qs. I find that notation a little confusing. Especially as a few lines down sediment load is denoted ïĄĎY.

Page 7, line 16: what is a locally made abacus and how does it work?

Page 8, line 14: A subdivision of what exactly?

Page 9, line 8: what flow frequencies are being referred to? Annual, daily, instantaneous, all of them?

Page 9, line 9: What is meant by 'irregular flow'

Page 9 line 10: No results for the exponential distribution are included in this study?

Section 3.7: I don't think this section is necessary

Page 11, line 8-11: I don't understand this sentence. What is QT99, and what is meant by '1% of the annual time'? Is this based on analysis of annual maximum data, or all daily flow data? Also, there is a reference to Fig.2 but I have no idea from the text what I am looking at in that Figure. More explanation is required here. Finally, this section used QY for sediment (check units in line 16, page 11) rather than QS as on page 3.

Page 12, line 3: From the description in the text I am not sure what I am looking at in Figure 3. Please try to be more helpful to the reader.

Section 4.2: This headline is not very helpful in describing what is the content of this section.

Page 17, line 8: Qs, but should that be QY?

Figures 7 and 8: The layout of these two figures is different and it would be better if they had a more uniform look. For example, remove gridlines from Figure 8, add y-axis label on Figure 7.

---

## Author Comment (AC1) · 30 Aug 2018

Replies by the authors

R1: The paper is interesting and well within the scope of the journal, nevertheless it needs to be reinforced with reference to several points, mainly regarding paper clarity and organization.

Reply: Thanks for the general comment and all your suggestions to makes the paper better organized and clearer.

R1: The title should more informative and let the reader know that it deals with sediment transport and hydrological implications.

Reply: We propose a new title: "Mean and analytical methods to characterize the efficiency of floods to move sediments in a small semi-arid basin". The mean method refers to the use of histograms where each class of discharge is represented by its mean value, and the analytical method in which the dominant discharge is defined as the solution of h'(Q)=0 where h(Q)= f(Q).g(Q), f(Q) being a probability function of the flow frequency and g(Q) a sediment rating curve. These names are used in the literature, see for example Crowder and Knapp (2005) and Lenzi et al. (2006) for the mean approach, and Nash et al. (2005), Goodwin (2004), Quader et al. (2008) and Bunte et al. (2014) for the analytical approach.

R1: The introduction should be clearer with respect to the paper's objectives and novelty. While it is clearly stated that three methods for individuation of discharge classes are compared, it is not evident how many and which methods are compared with reference to the evaluation of effective discharge, and other related methods including evaluation of return times.

Reply: We thoroughly revised the presentation of the paper's objectives in the following way (new summary):

Over a long multi-year period, flood events can be classified according to their effectiveness in moving sediments. Efficiency depends both on the magnitude and frequency with which events occur. The effective (or dominant) discharge is the water discharge which corresponds to the maximum sediment supply. If its calculation is well documented in temperate or humid climate and large basins, it is much more difficult in small and semi-arid basins which encompass short floods with high sediment supplies. On the example of 31-years of measurements in the Wadi Sebdou (North-West

Algeria), this paper compares the two main approaches to calculate the effective discharge (the mean approach based on histograms of sediment supply by discharge classes and an analytical calculation based on a hydrological probability distribution and on a sediment rating curve) to a very simple proxy: the half-load discharge, i.e. the flow rate corresponding to 50% of the cumulative sediment yield. Three types of discharge subdivisions were tested. In the mean approach, two subdivisions provided effective discharges close to the half-load discharge. Analytical solutions based on Log-normal and Log-Gumbel probability distributions were assessed but they highly underestimated the effective discharge, whatever the subdivision used to adjust the flow frequency distribution. Furthermore, annual series of maximum discharge and half-load discharge enabled to infer the return period of hydrological years with discharges higher than the effective discharge (around 2 years) and to show that more than half of the yearly sediment supply is carried by flows higher than the effective discharge only every 7 hydrological years. This study was the first to adapt the statistical approach in a semi-arid basin and to show the potentiality and limits of each method in a such climate.

The revised introduction has been rewritten accordingly, and the structure of the revised paper as well.

R1: While the paper is strongly focused on issue related to the representativeness of available measures (discharge and concentrations), only few explanations are provided about physical processes related to sediment transport. The eventual presence of hysteresis which is at the core of many works on the topic is here barely mentioned.

Reply: This paper focuses on statistical analysis of sediment discharge. We tried to state it more clearly in the revised title. Physical processes are included through their "signatures" in the data collected all along the 31-year period of measurements but this paper does not study the time variations of parameters at short term. Only histograms, statistics and yearly values are considered. A specific study of hysteresis during floods (so at short term) was already published on the same basin (see our paper: Megnounif
et al., 2013, Journal of Hydrology). A sentence was added in the site description accordingly: "Previous studies on sediment dynamics in this basin proposed syntheses on the main parameters, or on sediment processes at the origin of hysteresis phenomena during floods, based on the detailed analysis of short-term time variations of water and sediment discharges (Megnounif et al., 2013)."

R1: A figure representing the basin and its position is missing and basin description is almost entirely addressed to other papers referenced.

Reply: The section title 'study area' was changed to 'study area and data collection'. Its text was rewritten from former sections 2 and 3.3 (in reply to your next suggestion). A figure representing the watershed position was inserted.

R1: Also, the organization of section 3 Methodology, does not help the reader in understanding what the authors are mainly presenting and comparing. Making an exception for subsections 3.2 about class intervals, all other parameters are presented without any specif ic order of hierarchy. Subsection 3.3 about hydrometric measurements and pre-processing could be probably moved in section 2.

Reply: Section 3 was revised according to the new thread. Subsection 3.3 was separated into three subparts, two of which moved to the introduction and to the revised section 2. Other subsections of section 3 are now based on the mean approach, the analytical approach, the half-load discharge and the return periods.

R1: The evaluation of the Half-load discharge does not actually add any knowledge insight apart for a weak literature comparison.

Reply: As we show, the half-load discharge (29.8 m3 s-1) is a very good approximation of the effective discharge (either 29.5 or 29.01, depending on the discharge subdivision). Furthermore, it can be estimated very quickly from the dataset since it is directly readable from the cumulative sediment curve (Fig. 3), without any calculation. We thus propose to keep this indicator, which can be easily accessed for practical applications
by technical services or managers. The following sentence was added: "Its very quick and easy determination from the cumulative sediment yield curve makes it a suitable indicator for practical applications by technical staff or managers."

Furthermore, in the Wadi Sebdou, the half-load discharge in 1973-1988, QY50 (7.68 m3 s-1), was close to the dominant discharge QD (6.7 m3 s-1) and not far from the modal class [6.1; 7.4 m3 s-1[; in 1988-2003, QY50 (31.80 m3 s-1) was very close to the modal class [26.4; 31.7 m3 s-1[ whose center was defined as the effective discharge (QD = 29.0 m3 s-1). Thus, in the Sebdou Basin, the half-load discharge can be seen as a robust proxy for the effective discharge. This result fosters further warrants in future studies and in other basins. This was emphasized in the last paragraph of the discussion.

R1: The same applies to the subsection 3.7 Recurrence interval which is a mere evaluation of a certain discharge compared to the distribution of maximum and mean discharge.

Reply: The recurrence interval of a certain discharge like the effective discharge is traditionally calculated in hydrology. However, this return period is only based on hydrologic distributions (as you explain, from the distributions of mean and maximum discharge). We propose in this paper something more original, calculating a recurrence interval of the effective discharge compared to the distribution of annual half-load discharge, which was obtained from both water and sediment data time series, and investigate its additional information. The text was clarified accordingly.

R1: Figure 3, upper portion, should be placed in a different scale, data are almost invisible.

Reply: Thank you for the suggestion. We changed the upper portion in semi-log scale, so that data become more visible.

R1: Last sentence of page 17 is not clear, should be rephrased.

[Figure]

Reply: A subsection was entirely devoted to the return periods calculations. The last sentences of this subsection (4.2) were rephrased to: "The difference of nearly five years between these two estimates is attributed to their different meanings. While one indicates that the effective discharge is observed at least once in a hydrological year roughly every two years at the gauging station, the other shows that half of the yearly sediment supply is carried by flows higher than the effective discharge only every 7 hydrological years."

Just above, another sentence was modified to clarify this topic.

R1: The striking difference between analytical and statistical approaches is simply distressing and does not find a satisfactory justification.

Reply: The comparison of the two calculations makes it possible to add information on the limits of the analytical method (based on a probability density function for the flow frequency). It did not provide good results in semi-arid environments because the sediment rating curve introduces errors (designed as 'of the first type' in the paper) and because pronounced asymmetric probability distributions failed to reproduce good frequencies of high discharge (errors 'of second type'). Consequently, this comparison shows that the mean method by decomposition of histogram classes is the most suitable in a semi-arid environment. This had never been tested in the literature. It's an original result. This was emphasized in subsection 5.4.

R1: Figure 9c and 9d, are hard to understand. Maybe that placing indications of number (1) to (6) on the time series (i.e. in figures 9a and 9b) may help.

Reply: Figure 9 and the former subsection 4.3 on the characterization of flows and associated sediment loads were deleted, to strengthen the paper on its main purpose and its originality.

R1: The discussion section is quite long and not always add useful information. Some parts could be shortened and moved to other subsections, for the sake of paper structure and readability. Other parts may be even canceled like lines 1-6 at page 25 or lines 1-10 page 26 or the entire 5.5 subsection.

Reply: The discussion was revised and shortened. In particular, the former entire subsections 5.3 (former lines 1-10 page 26) and 5.5 were removed, as you requested. Globally, the paper was reduced by ∼10% (around 9700 words against 10700 in the previous version).

Please also note the supplement to this comment:
https://www.hydrol-earth-syst-sci-discuss.net/hess-2018-189/hess-2018-189-AC1-supplement.pdf

**Supplement:**

[revised manuscript text omitted]

---

## Author Comment (AC2) · 30 Aug 2018

Replies by the authors

R2: I think this is an interesting manuscript potentially reporting on an interesting data set and a thorough analysis. However, in my opinion the authors fail to make a convincing case for while this paper is a significant and original contribution to the scientific literature.

[Figure]

Reply: Thank you for your general appreciation and your detailed comments and suggestions to improve the paper. In particular, the revised version underlines the originality of our work.

R2: The introduction is mostly a methodological introduction. There is some text in section 3.3 which does try to describe the scientific context of this particular study which could perhaps be used as a starting point for a more focused introduction, reviewing the literature and identifying knowledge gaps.

Reply: Thank you. We thoroughly revised the thread of the paper's objectives in the following way (new summary):

Over a long multi-year period, flood events can be classified according to their effectiveness in moving sediments. Efficiency depends both on the magnitude and frequency with which events occur. The effective (or dominant) discharge is the water discharge which corresponds to the maximum sediment supply. If its calculation is well documented in temperate or humid climate and large basins, it is much more difficult in small and semi-arid basins which encompass short floods with high sediment supplies. On the example of 31-years of measurements in the Wadi Sebdou (North-West Algeria), this paper compares the two main approaches to calculate the effective discharge (the mean approach based on histograms of sediment supply by discharge classes and an analytical calculation based on a hydrological probability distribution and on a sediment rating curve) to a very simple proxy: the half-load discharge, i.e. the flow rate corresponding to 50% of the cumulative sediment yield. Three types of discharge subdivisions were tested. In the mean approach, two subdivisions provided effective discharges close to the half-load discharge. Analytical solutions based on Log-normal and Log-Gumbel probability distributions were assessed but they highly underestimated the effective discharge, whatever the subdivision used to adjust the flow frequency distribution. Furthermore, annual series of maximum discharge and half-load discharge enabled to infer the return period of hydrological years with discharges higher than the effective discharge (around 2 years) and to show that more

than half of the yearly sediment supply is carried by flows higher than the effective discharge only every 7 hydrological years. This study was the first to adapt the statistical approach in a semi-arid basin and to show the potentiality and limits of each method in a such climate.

The revised introduction has been rewritten accordingly, and the structure of the revised paper as well. In particular, the revised introduction now reviews the literature, identifies knowledge gaps and states more clearly the novelty of this paper. A part of the former section 3.3 was moved to the introduction, while some methodological information moved from the former introduction to the section 3 (such as the upper page 3 of the former version).

The title of the paper was also revised, as suggested by the referee #1, according to: "Mean and analytical methods to characterize the efficiency of floods to move sediments in a small semi-arid basin". The "mean" method refers to the use of histograms where each class of discharge is represented by its mean value, and the analytical method is such that the dominant discharge is the solution of $h'(Q)=0$ where $h(Q)= f(Q).g(Q)$, $f(Q)$ being a probability function of the flow frequency and $g(Q)$ a sediment rating curve. These names are used in the literature, see for example Crowder and Knapp (2005) and Lenzi et al. (2006) for the mean approach, and Nash et al. (2005), Goodwin (2004), Quader et al. (2008) and Bunte et al. (2014) for the analytical approach.

R2: Also, the manuscript is quite long as there is a lot of fundamental methodology included. I think it would be more readable if the focus was more on the original aspects of the analysis with less reference to standard methods used.

Reply: The revised version was focused on original aspects of the analysis and some former paragraphs or sections (like the former sections 4.3, 5.3 and 5.5) were removed.

R2: The conclusion is very long. I would suggest a more concise set of conclusions would help to communicate the potential importance of the paper to readers.

Reply: The conclusion was shortened and focused on original aspects of this work. Globally, the paper was reduced by ~10% (around 9700 words against 10700 in the previous version).

R2: In summary: this is potentially an interesting paper, but there is much that can be done in order to improve the quality of the presentation.

Other comments Section 3: Discharge is Q, concentration is C, and the product of the two is Qs. I find that notation a little confusing. Especially as a few lines down sediment load is denoted $\Delta$Y.

Reply: We used the traditional and most common annotation for Q (discharge) and the sediment flow. While subscript S stands for "sediment" discharge in Qs (weight or volume of transported sediment per unit of time), Y stands for sediment yield (in weight or volume of transported sediments), and QY$\alpha$ stands for a water discharge (and not a sediment discharge) corresponding to a cumulative sediment yield of $\alpha$ %. We used different names (and the most standard ones) for parameters of different units so as to avoid any confusion. However, you rightly pointed to a bad wording in the former subsection 3.1 when we referred to "inputs of sediment load" rather than elementary sediment yield, and "sediment yield Qs" instead of "sediment discharge Qs". The wording was corrected and double-checked.

R2: Page 7, line 16: what is a locally made abacus and how does it work?

Reply: In the ANRH protocol, the flow is generally measured with a winch by gauging a section over 5–8 verticals with between 2 and 6 measurements per vertical. At night, during holidays, or during some floods, the discharge is derived from a limnimetric height using a local stage-discharge relationship or abacus (see Achite and Ouillon, 2007, Journal of Hydrology). The local stage-discharge relationship or rating curve has been derived from the limnimetric heights and the river flows measured by the winch at the station, and is regularly updated. The word "abacus" was removed and replaced by a "stage-discharge relationship".

R2: Page 8, line 14: A subdivision of what exactly?

Reply: This paragraph refers to the choice of adjacent categories (or bins) of the discharge histogram. The title was revised into: "Relevance of a subdivision of discharges".

R2: Page 9, line 8: what flow frequencies are being referred to? Annual, daily, instantaneous, all of them?

Reply: This refers to instantaneous discharges measured at the gauging stations. However, a left skewed distribution may also be observed with other short-term (e.g. hourly, daily) discharge values. The sentence was completed following: "Probability density functions representing flow frequencies from instantaneous values are left skewed distributions"

R2: Page 9, line 9: What is meant by 'irregular flow'

Reply: This refers to semi-arid environments, where streams/wadis may encompass long periods of nil or very small discharge. The sentence was modified into: "However, for irregular flows as encountered in semi-arid environments with long low flow periods, more pronounced asymmetric distributions are recommended."

R2: Page 9 line 10: No results for the exponential distribution are included in this study?

Reply: The reference to the exponential solution was removed from the paper. To complete your information: we developed and calculated the statistical solution for the exponential distributions. The deriving discharge probability density function of flow frequencies was only acceptable when we used a subdivision of flow classes of equal length 6 or 8 m3/s. These subdivisions were, however, not retained in this paper since they do not check the selection criteria of the modal class. The subdivision of discharges into classes in geometric progression did not provide a suitable adjustment to the exponential distribution.

R2: Section 3.7: I don't think this section is necessary

Reply: This study showed that, in the Wadi Sebdou, the half-load discharge (29.8 m3/s) is a very good approximation of the effective discharge (either 29.5 or 29.01, depending on the discharge subdivision). Furthermore, it can be estimated very quickly from the dataset since it is directly readable from the cumulative sediment curve (Fig. 3), without any calculation. We thus propose to keep this indicator, which can be easily accessed for practical applications by technical services or managers. The following sentence was added: "Its very quick and easy determination from the cumulative sediment yield curve makes it a suitable indicator for practical applications by technical staff or managers."

Additionally, we showed in the last subsection of the discussion that the half-load discharge calculated over two hydrologic periods (1973-1988 and 1988-2033) was very close to the effective discharge of each period, making it a robust proxy of the effective discharge. This result fosters further warrants in future studies and in other basins. A short paragraph was added at the end of the discussion.

R2: Page 11, line 8-11: I don't understand this sentence. What is QT99, and what is meant by '1% of the annual time'? Is this based on analysis of annual maximum data, or all daily flow data? Also, there is a reference to Fig.2 but I have no idea from the text what I am looking at in that Figure. More explanation is required here.

Reply: The quantiles are presented at the end of the subsection "Elementary contribution and budgets". The instantaneous discharge is in average higher than QT99 during 1% of the time each year (i.e. 87 hours and 40 minutes). QT99 calculation is based on all elementary contributions of flow during 31 years of measurements (40,081 data, as detailed in the "Data pre-processing" subsection, i.e. around 4 values per day, in average). The link with the figure 3 (former Fig. 2) is the following: QT99 is directly readable on the curve of the cumulative time duration assigned to ordinal discharges; it is the abscissa of the cumulative curve when its ordinate is 99%. You can check as well on the figure that QT90=1.54 m3/s, which means that 90% of the year, in average, the instantaneous discharge is lower than QT90. The paragraph was rewritten.

R2: Finally, this section used QY for sediment (check units in line 16, page 11) rather than QS as on page 3.

Reply: QY refers to a water discharge (in m3 s-1, and such that QYx is the water discharge below which x % of the cumulative sediment yield was brought by the stream) while Qs refers to a sediment discharge (in mass of suspended matter per unit of time). Units in line 16 page 11 are thus good. However, your remark is very instructive, since the scientific community use alternatively QY50 or QY1/2 for the same parameter (the half-load discharge or the mean discharge in terms of sediment yield). To be consistent all along the paper with the subsection "Elementary contributions and budgets", QY50 (i.e. the water discharge that delimits 50% of the cumulative sediment yield) was preferred to QY1/2 in the revised version of the paper.

R2: Page 12, line 3: From the description in the text I am not sure what I am looking at in Figure 3. Please try to be more helpful to the reader.

Reply: Thank you for this remark. We changed the second sentence according to: "As can be seen on the histogram of sediment yields (Fig. 4), the class which induced the highest sediment contribution (the dominant class), [29; 30 m3 s-1[, brought 4.8% of the total sediment supply. This class represents 0.51% of the total water supply (Fig. 4) [. . .]"

R2: Section 4.2: This headline is not very helpful in describing what is the content of this section.

Reply: The headline was changed and replaced by: "Analytical determination of the effective discharge" (title of 4.3 in the revised paper)

R2: Page 17, line 8: Qs, but should that be QY?

Reply: While the calculation of the effective discharge by the mean approach makes use of the elementary contributions of sediment supply (thus introducing QY), the analytical approach makes use of the probability distribution of instant parameters. We

referred to the Wolman and Miller's presentation of the analytical method who introduced g(Q), the rating curve estimating the suspended sediment flux Qs as a function of the water discharge (see Introduction and Fig. 1).

R2: Figures 7 and 8: The layout of these two figures is different and it would be better if they had a more uniform look. For example, remove gridlines from Figure 8, add y-axis label on Figure 7.

Reply: Done.

Please also note the supplement to this comment:
https://www.hydrol-earth-syst-sci-discuss.net/hess-2018-189/hess-2018-189-AC2-supplement.pdf

**Supplement:**

[revised manuscript text omitted]

---

## Author Response (AR2)

**Submission of the revised version of the paper originally entitled "How to determine the effective discharge and its return period in a semi-arid basin? The case of the Wadi Sebdou, Algeria (1973–2004)" by Abdesselam Megnounif and Sylvain Ouillon**

11 Nov 18 :
We kindly ask you to revise your manuscript accordingly and to upload the revised files, a point-by-point reply to the comments, and a marked-up manuscript version showing the changes made in your File Manager no later than 21 Nov 2018: https://editor.copernicus.org/HESS/file_manager/hess-2018-189.

**Editor Decision: Publish subject to minor revisions (review by editor)** (11 Nov 2018) by Thomas Kjeldsen

Comments to the Author:

Very minor corrections required by one reviewers. Otherwise, both reviewers are satisfied that the revision answers the questions raised during the first review.

**Submitted on 28 Sep 2018**

**Anonymous Referee #1**

*Suggestions for revision or reasons for rejection (will be published if the paper is accepted for final publication)*

The paper has been much improved after the authors revisions and now the methodology, results and conclusions are more clearly stated. The author addressed point by point the first review comments. The new title of the paper is more clear than the old one but I still have some concern about it because, even considering the authors' explanations, "Mean and analytical methods" sounds weird. As it is, it needs Italic format or quotation marks to avoid confusion. Maybe the old version "empirical and statistical" is better for the first part of the title.

**Submitted on 09 Nov 2018**

**Anonymous Referee #2**

*Suggestions for revision or reasons for rejection (will be published if the paper is accepted for final publication)*

This is a revised version of the manuscript. The authors have done a thorough job of revising the paper and have addressed most of my initial concerns. A few minor comments:

Not sure I like the new title as it is not immediately obvious what 'Mean' refers to.

Figure 8: no y-axis label, and also probabilities are more commonly quoted as numbers between 0 and

5  1.

**Reply to the editor (20 November 2018)**

Dear editor,

10  We would like to thank again the two reviewers for their suggestions to improve the paper. It was revised accordingly on two points: (1) as suggested, we removed "the *mean* method" which was confusing and proposed to revise the title following "Empirical and analytical methods…"; (2) on Fig. 8 and Fig. 9, the y-axis was changed from 0-100% to 0-1 and a y-axis title was added.

Additionally, few small changes were brought. In particular, another affiliation was added for the second

15  author and acknowledgements were added.

Thank you again for editing this paper. Best regards,

Sylvain Ouillon and Abdesselam Megnounif

[revised manuscript text omitted]